# A structure-based mRNA vaccine for Nipah virus in healthy adults: a phase 1 trial

Nipah virus (NiV) is a highly pathogenic, zoonotic paramyxovirus with pandemic potential. No licensed vaccines or treatments are available. Therefore, we conducted a phase 1, first-in-human, open-label, dose-escalation trial of a lipid nanoparticle mRNA vaccine, mRNA-1215, encoding a chimeric pre-fusion F (Pre-F) protein linked to glycoprotein G of a NiV Malaysian strain. Forty healthy adults, who met eligibility criteria, were enrolled into the 10-, 25-, 50- or 100-µg dose groups with ten participants per group. Each participant received two doses of mRNA-1215, intramuscularly, at a 4-week interval except for one participant in the 10-µg dose group who received only the first dose. All participants remained in the study until their final study visit. The primary outcome of vaccine safety and tolerability was determined by prespecified end points: the frequency and severity of solicited local and systemic adverse events (AEs), unsolicited AEs, safety laboratory measures, medically attended AEs, AEs of special interest, new chronic medical conditions and serious AEs. The most frequently reported local and systemic AEs were mild pain and tenderness at the injection site ($n$ = 33; 82%) and mild malaise ($n$ = 16; 40%), respectively. Overall, the vaccine was well tolerated. No serious AEs occurred during the study. mRNA-1215 elicited robust Pre-F and G binding antibody titers, the prespecified secondary end points. An exploratory analysis found that mRNA-1215 elicited neutralizing titers by 2 weeks after the prime in all dose groups. Responses increased after the boost and remained elevated for at least 1 year after vaccination. These findings demonstrate an initial overall favorable safety profile and immunogenicity results for a first-in-human, structure-based, chimeric mRNA Nipah virus vaccine. mRNA-1215 is a promising vaccine candidate for continued clinical development for populations at risk for regional and potentially larger outbreaks caused by Nipah virus. ClinicalTrials.gov identifier: NCT05398796.

Nipah virus (NiV), a member of the *Paramyxoviridae* virus family, poses a notable risk to public health due to its pandemic potential and the absence of a licensed vaccine or therapeutic to prevent or mitigate severe disease or death. This virus is a zoonotic pathogen initially discovered in 1998 as the cause of an outbreak of viral encephalitis among pig farmers in Malaysia[1,2]. Since then, NiV has caused sporadic human outbreaks of respiratory illness and encephalitis in Singapore, India, the Philippines, and almost annual outbreaks in Bangladesh since 2001.

Most cases involve animal-to-human transmission; however, cases of human-to-human transmission have also been found[3,4]. Overall, for the nearly 30 NiV outbreaks reported so far, case fatality rates have ranged from 40% to 75% (ref. 5).

Fruit bats of the *Pteropodidae* family, genus *Pteropus*, are the natural host of NiV, and human infection results from direct contact with bat excretions or through indirect transmission via an intermediate host, such as pigs or horses. The geographical range of the *Pteropodidae*

✉ e-mail: dropulicl@niaid.nih.gov

**Fig. 1 | CONSORT diagram.** Forty healthy adults aged 18–60 years were enrolled between 11 July 2022 and 22 August 2023 to receive Nipah virus mRNA-1215 vaccine administered as a prime-boost regimen at weeks 0 and 4. *Of the 42 ineligible participants, three each met two exclusion criteria.

‡Participant discontinued study product administration after the first dose based on the Investigational New Drug (IND) sponsor/Principal Investigator decision but completed follow-up visits. †Participant's final study visit occurred at visit 16 (day 224) due to moving from the area. i.m., intramuscular.

bat family is extensive and covers Indo-Pacific territories across South and Southeast Asia, the Western Pacific regions, and much of Sub-Saharan Africa, in which over 2 billion people reside[6]. Henipa-like viruses have also been detected in several bat species in Brazil[7]. The high mortality rate associated with almost yearly outbreaks, evidence of human-to-human transmission, and the extensive geographical range of the natural host set the stage for a potential, future NiV epidemic or pandemic. As such, leading public health organizations, including the World Health Organization and the Coalition for Epidemic Preparedness Innovations, list NiV as a high priority pathogen and declare an urgent need to develop safe and effective vaccines for potential rapid deployment in affected countries[8–10].

Thus far, vaccine development against NiV has focused on the G glycoprotein, which mediates viral attachment to the host cell receptor, Ephrin B2/B3, as the vaccine antigen[11,12]. Preclinical animal studies using soluble proteins or vectored vaccines expressing the G protein demonstrated elicitation of neutralizing responses capable of blocking virus entry[13–20]. Several G protein vaccines have progressed to evaluation in phase 1 clinical trials, including a live-attenuated, recombinant vesicular stomatitis virus vaccine (PHVO2) expressing both the G glycoprotein of NiV Bangladesh strain (NiV(B)) and the Zaire Ebola virus glycoprotein (NCT05178901; NCT06221813); a replication-deficient simian recombinant adenovirus vector vaccine encoding NiV(B) G, ChAdOx1 NiV (ISRCTN87634044); and HeV-sG-V, a recombinant protein subunit vaccine containing a portion of the G glycoprotein of Hendra virus (HeV), a closely related henipavirus (NCT04199169). Building on these efforts, we incorporated an additional viral antigen in our NiV vaccine[21,22].

The F glycoprotein is the second key NiV surface glycoprotein that drives viral entry by fusing the viral membrane to the host cell membrane upon attachment of NiV-G to the Ephrin B2/B3 receptor. This fusion event is driven by a conformational change of the F protein from a metastable pre-fusion (Pre-F) conformation to a stable, post-fusion (Post-F) form. Vaccination of mice with a mutation-stabilized Pre-F protein generated greater binding and neutralizing antibody titers compared to vaccination with the Post-F protein, suggesting that the immune response to Pre-F can inhibit the virus fusion machinery, and thereby inhibit infection of cells in vitro[21–23]. Similar results were observed with other vaccines, such as the respiratory syncytial virus (RSV) vaccine, which is now commercially available[24]. Thus, including

the stabilized Pre-F protein as an antigen in a NiV vaccine should enhance its immunogenicity and efficacy. Additionally, amino acid sequence alignments across multiple henipaviruses show that the F protein is more conserved than the G protein, suggesting that a vaccine containing F may elicit an immune response with greater breadth[25]. Thus, neutralizing antibodies targeting highly conserved epitopes of the Pre-F protein may exhibit better cross-reactivity against other henipaviruses compared to antibodies targeting the G protein[23]. For these reasons, we developed a messenger RNA vaccine against NiV that expresses both the stabilized Pre-F and G glycoproteins (Pre-F/G) of a NiV Malaysian strain (NiV(M)).

Mice vaccinated with mRNA vaccines encoding either chimeric Pre-F/G, Pre-F or Post-F induced robust F-specific binding antibody responses; however, only mice vaccinated with Pre-F/G or Pre-F mRNA generated neutralizing antibody responses[22]. Vaccination with stabilized Pre-F/G mRNA or with a dimer of G trimers (hexameric G) mRNA resulted in similar G-specific-binding antibody titers and equivalent neutralization titers. In contrast, the Pre-F/G mRNA vaccine elicited superior NiV-specific CD4+ and CD8+ T cell responses, as well as T follicular helper (T_FH) cell responses to both Pre-F and G[22]. These preclinical, proof-of-concept studies demonstrated that including both Pre-F and G proteins as vaccine antigens enhances the breadth of both humoral and T cell-mediated immune responses, making the chimeric Pre-F/G mRNA vaccine the lead candidate for clinical development. Here, we report the results of a phase 1, first-in-human, dose-escalation study that evaluated the safety, tolerability and immunogenicity of a two-dose vaccination regimen of the chimeric Pre-F/G mRNA vaccine, mRNA-1215, in healthy adult volunteers.

## Results

### Trial participants

A total of 40 participants, 18 females (45%) and 22 males (55%), with an overall mean age of 37 years (range 22–59) were enrolled into the study from 11 July 2022 to 22 August 2023 (Fig. 1 and Table 1). Eligible participants were adults 18–60 years of age who were in good general health as determined by medical history, physical examination and laboratory testing. Specific exclusion criteria related to NiV included confirmed past NiV infection or previous residence for more than 6 months or planned travel during the study for any length of time to places where NiV infection is endemic. The full list of eligibility criteria

**Table 1 | Demographic characteristics of study participants**

| | Dose of mRNA-1215 | | | | Overall |
|---|---|---|---|---|---|
| | 10 µg (n=10) | 25 µg (n=10) | 50 µg (n=10) | 100 µg (n=10) | (n=40) |
| **Sex[a], n (%)** | | | | | |
| Male | 5 (50.0%) | 8 (80.0%) | 4 (40.0%) | 5 (50.0%) | 22 (55.0%) |
| Female | 5 (50.0%) | 2 (20.0%) | 6 (60.0%) | 5 (50.0%) | 18 (45.0%) |
| **Age[b], years** | | | | | |
| Mean±s.d. | 38.5±11.1 | 33.9±6.5 | 40.9±11.1 | 36.2±13.1 | 37.4±10.6 |
| Range | 22.0–49.0 | 23.0–44.0 | 23.0–58.0 | 23.0–59.0 | 22.0–59.0 |
| **Race, n (%)** | | | | | |
| Asian | 0 (0.0%) | 2 (20.0%) | 3 (30.0%) | 1 (10.0%) | 6 (15.0%) |
| Black or African American | 2 (20.0%) | 1 (10.0%) | 1 (10.0%) | 1 (10.0%) | 5 (12.5%) |
| White | 8 (80.0%) | 7 (70.0%) | 6 (60.0%) | 8 (80.0%) | 29 (72.5%) |
| **Ethnicity, n (%)** | | | | | |
| Non-Hispanic/Latino | 9 (90.0%) | 10 (100.0%) | 9 (90.0%) | 10 (100.0%) | 38 (95.0%) |
| Hispanic/Latino | 1 (10.0%) | 0 (0.0%) | 1 (10.0%) | 0 (0.0%) | 2 (5.0%) |
| **Body mass index[c], kg m⁻²** | | | | | |
| Mean±s.d. | 27.1±4.4 | 27.4±2.9 | 25.4±3.5 | 26.2±4.1 | 26.5±3.7 |
| Range | 20.9–32.5 | 22.5–31.7 | 21.3–31.8 | 20.2–31.5 | 20.2–32.5 |
| **Education, n (%)** | | | | | |
| High school graduate/GED[d] | 0 (0.0%) | 0 (0.0%) | 1 (10.0%) | 0 (0.0%) | 1 (2.5%) |
| College/university | 5 (50.0%) | 3 (30.0%) | 6 (60.0%) | 5 (50.0%) | 19 (47.5%) |
| Advanced degree | 5 (50.0%) | 7 (70.0%) | 3 (30.0%) | 5 (50.0%) | 20 (50.0%) |

[a]Sex, race, and ethnic group were self-reported by the participants. [b]Age represents age at enrollment day. [c]Body mass index at enrollment. [d]GED is the General Educational Development high-school equivalency exam that certifies knowledge equivalent to a U.S. high school diploma.

can be accessed in the Methods and in the protocol (available in the Supplementary information and in ClinicalTrials.gov NCT05398796). Ten participants each in the 25-µg and 100-µg dose groups received two mRNA-1215 administrations, and 19 of 20 participants completed all follow-up visits. One participant in the 25-µg group missed a single visit. The ten participants in the 50 µg group completed both vaccine administrations, and all but one participant completed all follow-up visits. One participant in the 10 µg dose group received only the first dose of mRNA-1215 but completed all follow-up visits; the other nine participants in this group each received two vaccine administrations and completed all follow-up visits. The last study visit occurred on 17 September 2024.

The primary outcome of safety and tolerability and the secondary immunogenicity outcome were prespecified in the study protocol, and all outcomes are presented in this report.

## Vaccine safety
The primary objective of the trial was to evaluate the safety and tolerability of the mRNA-1215 vaccine administered intramuscularly as a two-dose regimen with a 4-week interval. The primary safety end points were: the frequency and severity of solicited local and systemic reactogenicity symptoms that occurred in the 7 days after each vaccination; the frequency and severity of unsolicited adverse events (AEs), including abnormal laboratory values during the 28 days after each vaccination; the frequency of serious AEs, AEs of special interest, AEs leading to withdrawal, or new chronic medical conditions occurring through the last study visit; and the frequency of medically attended AEs through 6 months after the last vaccination. Local reactogenicity symptoms were predominantly mild in severity and similar between the prime and booster doses (Fig. 2a). Mild pain and/or tenderness were the most frequently reported local symptom (n = 33 of 40, 83%) and

tended to occur more frequently (reported by 70–100% of recipients) in the 25-, 50- and 100-µg dose groups compared to the 10-µg group (reported by 33–50% of recipients; (Fig. 2a). Only two of these events extended beyond the 7-day reactogenicity period with one event lasting up to 11 days. Two participants in the 100-µg dose group reported local events of higher severity; severe redness (n = 1 of 40, 2.5%) and moderate pain (n = 1 of 40, 2.5%) occurred after the prime or booster dose administration, respectively (Fig. 2a).

Overall, half of the participants (n = 20 of 40, 50%) reported mild-to-moderate systemic reactogenicity symptoms after either the prime or booster dose (Fig. 2b). Malaise was the most common symptom reported after both the prime (n = 13 of 40, 33%) and booster dose administrations (n = 15 of 40, 39%) followed by headache (n = 14 of 40, 35%) and myalgia (n = 12 of 40, 30%) (Fig. 2b). There was a trend of a greater frequency and severity of systemic symptoms with an increase in vaccine dose and after the second dose. For any systemic symptom and for both vaccinations, three participants in the 10-µg dose group (n = 3 of 10, 30%) reported mild (n = 2) and moderate (n = 1) symptoms; four participants in the 25-µg dose group (n = 4 of 10, 40%) reported only mild symptoms (n = 4); six participants in the 50-µg dose group (6 of 10, 60%) reported mild (n = 5) or moderate (n = 1) symptoms; and eight participants in the 100-µg dose group (n = 8 of 10, 80%) reported mild (n = 4), moderate (n = 3) or severe (n = 1) symptoms (Fig. 2b). Five participants in either the 10-, 50- or 100-µg dose groups experienced mild or moderate headache or malaise that persisted beyond the 7-day reactogenicity period with events lasting 8–12 days.

Ten mild-to-moderate unsolicited AEs assessed as related to vaccination occurred in nine participants. One participant experienced two events (Table 2). The most frequently reported related AE was leukopenia (n = 4 of 40, 10%) after either the first (n = 3) or second (n = 1) vaccination and was most common in the 100-µg dose group

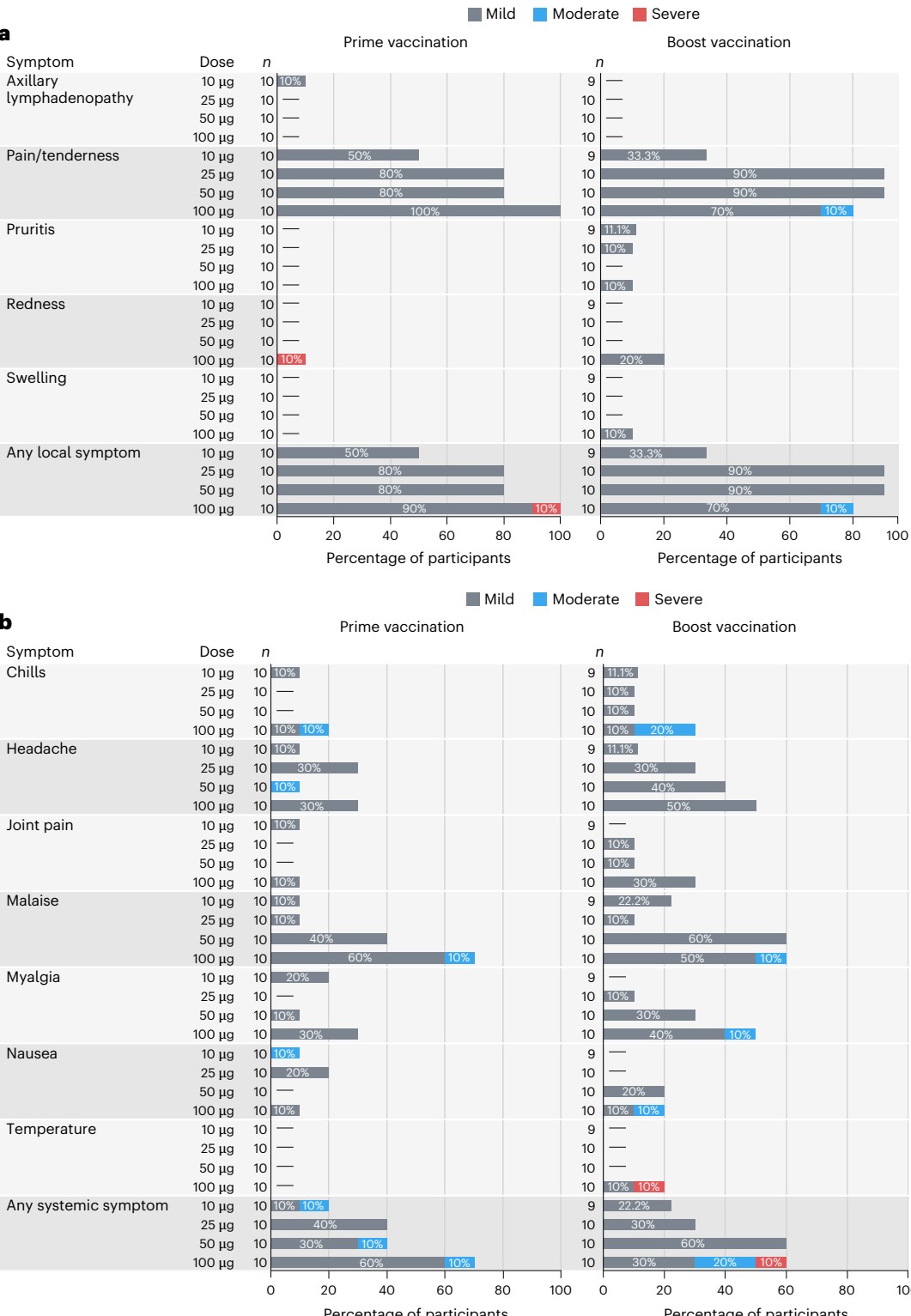

**Fig. 2 | Summary of maximum local and systemic reactogenicity following Nipah virus mRNA-1215 prime-boost vaccination. a,b,** Percent of participants (*x* axis) reporting any solicited local (**a**) or systemic (**b**) symptom (*y* axis) by dose groups in the 7 days following prime and boost doses. Results are shown for all (*n* = 40) study participants after the prime vaccination and for all but one participant, who did not receive a boost vaccination, in the 10-µg dose group (*n* = 39).

(*n* = 2). The leukopenia events were mild in severity and detected at 7 days after vaccination with three events resolving by 7 days and one resolving by 22 days. Skin and subcutaneous tissue disorders were the second most common category of related AEs reported (*n* = 3 of 40, 8%) (Table 2). Two (*n* = 2 of 40, 5%) participants who received the 10-µg dose developed an erythematous axillary rash, ipsilateral to the vaccination site, after the first dose. For one participant, the rash occurred within hours after vaccination and resolved in 19 days, and for the second participant, the rash occurred 10 days after vaccination and resolved in 6 days. The third participant was diagnosed with a RSV

**Table 2 | Unsolicited adverse events related to mRNA-1215 vaccination**

| | n | Dose of mRNA-1215 | Severity | Vaccination | Days after vaccination | Days to resolution |
|---|---|---|---|---|---|---|
| **Blood and lymphatic system disorders** | | | | | | |
| Leukopenia | 1 | 10 µg | Mild | Prime | 7 | 7 |
| | 1 | 50 µg | Mild | Prime | 7 | 7 |
| | 1 | 100 µg | Mild | Prime | 7 | 22 |
| | 1 | 100 µg | Mild | Boost | 7 | 7 |
| **Skin and subcutaneous tissue disorders** | | | | | | |
| Rash | 1 | 10 µg[a] | Mild | Prime | 0 | 19 |
| | 1 | 10 µg | Mild | Prime | 10 | 6 |
| Urticaria | 1 | 100 µg | Moderate | Prime | 12 | 198 |
| **Gastrointestinal disorders** | | | | | | |
| Diarrhea | 1 | 10 µg[a] | Moderate | Prime | 2 | 0 |
| **Vascular disorders** | | | | | | |
| Hypertension | 1 | 100 µg | Mild | Prime | 0 | 7 |
| **Nervous system disorders** | | | | | | |
| Dizziness | 1 | 100 µg | Moderate | Boost | 0 | 1 |

[a]The same participant experienced both events.

infection 2 days after receiving the first 100-µg dose. Ten days later, this participant developed urticaria. The urticaria worsened after the second dose and persisted for almost 6 months. The participant was treated with long-acting antihistamines for approximately 5 months, and medications were discontinued after the urticaria had resolved. Other AEs assessed as related to vaccination were moderate diarrhea ($n = 1$, 10-µg dose group); mild hypertension ($n = 1$, 100-µg dose group); and moderate dizziness ($n = 1$, 100-µg dose group) (Table 2); these events resolved on the same day, 7 days and 1 day after onset, respectively. No serious AEs, AEs of special interest, new chronic medical conditions lasting more than 1 year, participant withdrawal related to reactogenicity or to other AEs, or safety-related study pauses occurred during the trial.

## Vaccine-induced binding antibody responses to NiV(M) Pre-F and G

The secondary immunogenicity end point of this trial was to evaluate NiV-specific antibody responses at 2 weeks after the booster dose of mRNA-1215 administered as 10, 25, 50 or 100 µg. The mRNA-1215 vaccine was immunogenic at all doses. Both Pre-F and G glycoprotein-specific binding antibody responses had similar trends following the prime and boost vaccinations (Fig. 3a–d). Binding antibody IgG geometric mean titers to Pre-F and G reported as international units per ml (IU ml$^{-1}$) were detected as early as 2 weeks after the priming dose in all dose groups, with all participants displaying antigen-specific responses by 4 weeks after the first dose (Fig. 3a–d). Boost vaccination increased binding titers in each dose group, peaking 2 weeks after the boost (week 6); titers were uniformly elevated in all groups at this time point (Fig. 3a–d and Extended Data Tables 1 and 2). A gradual decline in Pre-F and G antigen-specific responses was observed over time; however, titers remained elevated well above baseline levels through the final study time point (week 56) for all dose groups (Fig. 3a–d and Extended Data Tables 1 and 2). Dose-dependent differences in both Pre-F and G titers were observed starting at week 24, 5 months after the boost, and at week 56, with lower titers in the 10-µg dose group compared to the other dose groups (Fig. 3c,d, Extended Data Tables 1 and 2 and Supplementary Tables 1 and 2). Anti-Pre-F and G titers 52 weeks post-boost were similar to titers at weeks 2–4 for all dose groups. (Fig. 3a,b and Extended Data Tables 1 and 2). We also performed a post hoc analysis using the longitudinal time course. Overall, the results

were consistent with our original standard analysis, except at week 6 we found that the 50-µg dose group displayed increased anti-Pre-F and anti-Mono-G titers when compared to the 10-µg dose group.

## Neutralizing antibody responses to NiV(M)

Neutralizing antibody responses following mRNA-1215 prime and boost vaccination, conducted as an exploratory analysis, had similar kinetics to the binding antibody responses. Neutralizing titers in IU ml$^{-1}$ were detected as early as 2 weeks after the first dose of mRNA-1215 (Fig. 3e,f) and increased after the second dose, with no difference in peak titers between dose groups 2 weeks after the boost (week 6) (Fig. 3e,f and Extended Data Table 3). After the peak, neutralizing titers gradually declined but remained above baseline through the final study time point (week 56) and were similar to titers at weeks 2–4 for all dose groups (Fig. 3e,f and Extended Data Table 3). As observed with the binding antibody titers, the only difference between the dose groups occurred at the week 24 and 56 time points with the 10 µg dose group having lower titers compared to each higher-dose group (Fig. 3f, Extended Data Table 3 and Supplementary Table 3).

## B cell responses to NiV(M) and cross-reactive B cell and neutralizing antibody responses to NiV(B)

Next, as an exploratory analysis, we evaluated the ability of mRNA-1215 to induce NiV(M) specific memory B cell responses and whether these responses cross-react with the closely related NiV(B) strain (amino acid identity percentage of 98.7% for F and 95.5% for G). We used a B cell probe binding assay with two different sets of protein probes, NiV(M) or NiV(B) Pre-F and G trimer probes (Fig. 4a,b and Extended Data Fig. 1). For both Pre-F and G proteins, a similar pattern was observed with most memory B cells binding to both NiV(M) and NiV(B) proteins. An increased frequency of antigen-specific memory B cells was detected at week 6 (2 weeks after the boost), and the frequency continued to rise in some participants in the 25-, 50- and 100-µg dose groups until week 56. A dose-response effect was observed with the 10-µg dose eliciting a lower B cell response compared to the higher-dose groups. A low frequency of B cells binding only to the NiV(M) G probe (but not only to the NiV(M) F probe) was detected in each dose group with an increase from baseline first detected at week 6 (Fig. 4b). This is consistent with the lower amino acid sequence similarity between NiV(M) and NiV(B) strains for G than F.

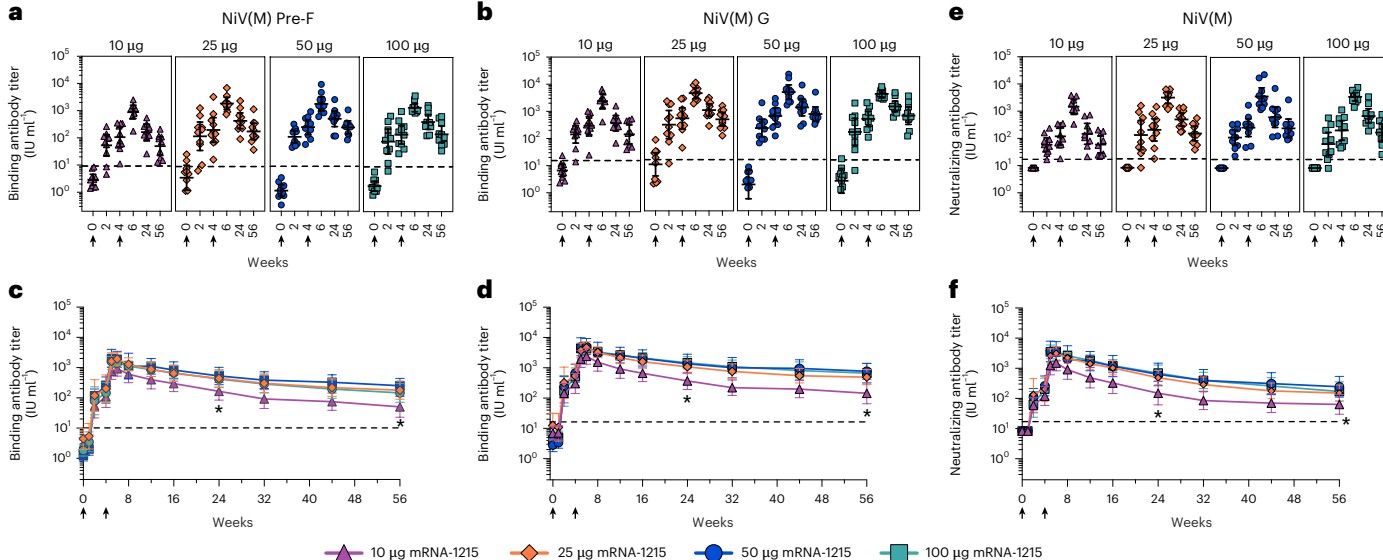

**Fig. 3 | Vaccine-induced binding and neutralizing antibody titers against NiV(M). a–f**, Binding antibody titers against Pre-F (**a**,**c**) and G antigens (**b**,**d**) were measured by ELISA and neutralizing antibody titers (**e**,**f**) were assessed by pseudovirus neutralization assay. Individual participant titers (geometric shapes) at selected time points with group geometric mean titers (GMTs) indicated by horizontal lines and 95% CIs as error bars (**a**,**b**,**e**). GMTs over time (geometric shapes) with 95% CIs (error bars) (**c**,**d**,**f**). Geometric shapes, purple triangles, orange diamonds, blue circles and teal squares, represent 10-, 25-, 50- and 100-μg dose groups, respectively. Titers are normalized to the Nipah virus antibody international standard and reported in IU ml$^{-1}$. Asterisks represent statistically significant differences between the GMTs of the 10 μg and the 25-, 50- or 100-μg dose group at week 24 (binding antibody titer: Pre-F $P$ value ≤ 0.032, G $P$ ≤ 0.007; neutralizing antibody titer $P$ ≤ 0.014) and week 56 (binding antibody titer: Pre-F $P$ ≤ 0.019, G $P$ ≤ 0.007; neutralizing antibody titer $P$ ≤ 0.029) determined by analysis of variance (ANOVA) and uncorrected pairwise two-sided $t$-test; adjustments for multiple comparisons were not performed. Black vertical arrows represent mRNA-1215 vaccine administration time points (week 0 and week 4). Dotted horizontal lines indicate lower limit of quantification (LLOQ) (**a**–**d**) and the limit of detection (LOD) (**e**,**f**). Results in **a**–**f** are shown for $n = 10$ participants in each dose group. One participant in the 10-μg group received only the first mRNA-1215 dose and data only from weeks 0, 2 and 4 are shown. The exact $P$ values for all between-group comparisons are presented in Supplementary Tables 1–3.

To demonstrate that mRNA-1215 could generate functional cross-reactive responses against NiV(B), serum samples from vaccine recipients were analyzed for neutralization of NiV F/G Bangladesh 2004 pseudo-typed virus (Fig. 4c,d and Extended Data Table 4). NiV(B)-specific neutralizing antibody titers followed a similar pattern as the NiV(M) responses and were detected by week 4, reaching peak levels by week 6. Week-56 titers remained elevated above baseline and were comparable to those at week 4 across all groups. Titers in the 10-μg dose group differed from all higher-dose groups only at week 24 (Fig. 4c,d and Supplementary Table 4).

### T cell responses against NiV(M)
Antigen-specific T cell responses were evaluated by intracellular cytokine staining in response to NiV(M) Pre-F or G peptide stimulation, as an exploratory analysis. Vaccination with mRNA-1215 elicited a type 1 helper T (T$_H$1) cell-dominant memory CD4$^+$ T cell response and a T$_{FH}$ cell response for each antigen that could be detected 2 weeks after the boost in each dose group (Extended Data Figs. 2 and 3). The highest frequency of antigen-specific T cells occurred at week 6 for each dose group, and were comparable between the groups. The vaccine also elicited a low frequency of NiV-specific memory CD8$^+$ T cells primarily in response to the Pre-F component of the vaccine antigen and only for doses above 10 μg.

### Cross-reactive binding and neutralizing antibody responses to HeV
To further characterize the breadth of mRNA-1215-induced immune responses, we conducted an exploratory analysis of binding and neutralizing antibody responses to HeV F and G antigens. The mRNA-1215 vaccine induced cross-binding antibodies to both HeV Pre-F and G proteins, which were detected 4 weeks after the first dose and increased up to 2 weeks after the boost (week 6; Extended Data Fig. 4a–d and Extended Data Table 5). For all dose groups, Pre-F and G antibody binding titers remained elevated above baseline through 56 weeks and were similar to Pre-F and G titers at week 4 (week 56 anti-G titers were different only for the 25-μg group). Starting at week 12, titers in the 10-μg dose group were slightly lower than each higher-dose group through week 56 (Extended Data Fig. 4c,d, Extended Data Table 5 and Supplementary Tables 5 and 6). mRNA-1215 vaccination also induced cross-reactive neutralizing antibody responses to HeV that were detectable 4 weeks after the prime in all dose groups with an increase in ID$_{50}$ titers after the boost. For all dose groups, the neutralizing antibody titers remained elevated through week 56 (Extended Data Fig. 4e,f, Extended Data Table 6 and Supplementary Table 7).

## Discussion
We report the results of a first-in-human phase 1, dose-escalation, open-label clinical trial that evaluated an mRNA vaccine that encodes a chimeric Pre-F and G NiV(M) vaccine antigen. This study focused on the initial evaluation of the safety, tolerability and immunogenicity of mRNA-1215. The vaccine was safe and well tolerated by 40 healthy adult volunteers, who experienced primarily mild-to-moderate reactogenicity. There were no safety-related study pauses or SAEs. mRNA-1215 was immunogenic in all participants and elicited robust antigen-specific binding and neutralizing antibody responses in the majority of participants as early as 2 weeks after the first dose. The magnitude of the binding and neutralizing responses was comparable between the dose groups through 2 weeks after the boost (the peak immune response). Although recipients of the 10-μg dose exhibited lower responses starting 5 months after the boost, the responses persisted and remained above baseline through 1 year. Antigen-specific T cell responses were also elicited with dominant CD4$^+$ T$_H$1 and T$_{FH}$ cell responses to both Pre-F

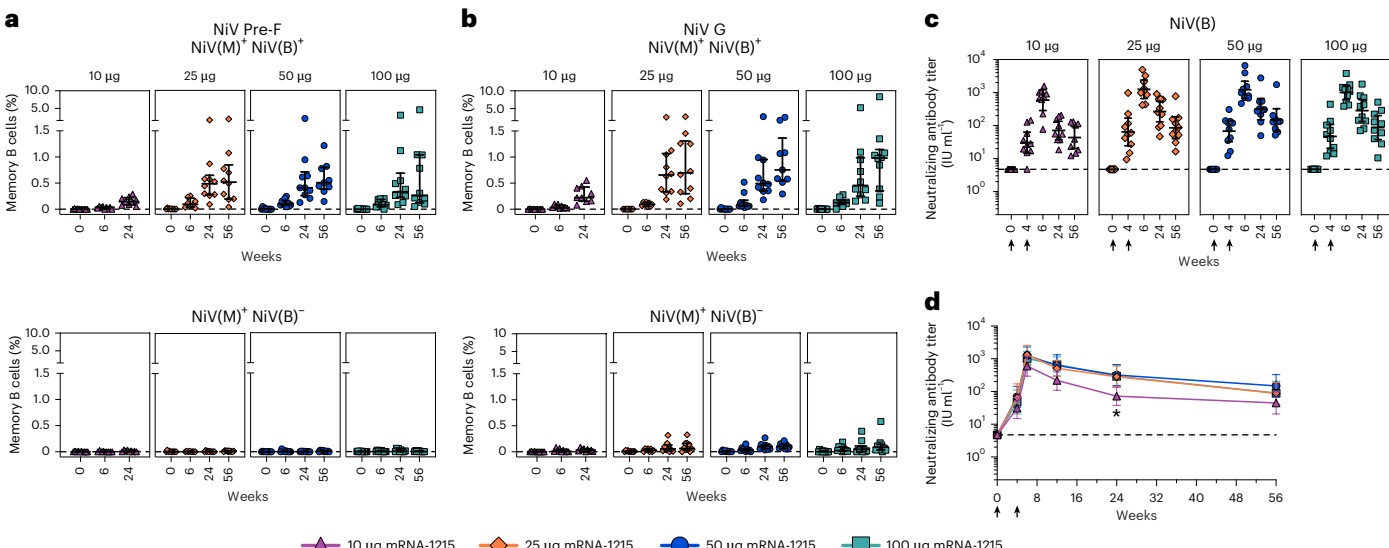

**Fig. 4 | Cross-reactive B cell responses and neutralizing antibody responses to NiV(B) antigen. a,b,** Frequencies of antigen-specific memory B cell responses were measured by staining of peripheral blood mononuclear cells with fluorescently labeled Pre-F and G trimer probe pairs matched for NiV(M) and NiV(B) at selected time points. Geometric shapes represent individual participant data; horizontal lines indicate median percentages and error bars show interquartile ranges. Top panels represent percentages of cross-reactive memory B cells specific for both NiV(M) and NiV(B) antigens; bottom panels show B cell responses specific for NiV(M) antigens only. **c,d,** NiV(B) neutralizing antibody titers were assessed by pseudovirus neutralization assay at selected time points (**c**) with group GMTs indicated by horizontal lines and 95% CIs as error bars and as a time course (**d**) displayed as GMTs (geometric shapes) with 95% CIs (error bars). Geometric shapes, purple triangles, orange diamonds, blue circles and teal squares, represent the 10-, 25-, 50- and 100-µg dose groups, respectively. Titers were normalized to the Nipah virus antibody international standard and reported in IU ml⁻¹. Asterisks represent statistically significant differences between the GMTs of the 10-µg and the 25-, 50- or 100-µg dose groups determined by ANOVA and uncorrected pairwise two-sided *t*-tests at week 24, *P* value ≤ 0.006; adjustments for multiple comparisons were not performed. Results in **a**–**d** are shown for *n* = 10 participants in each dose group. One participant in the 10-µg group received only a single dose of mRNA-1215 and data only from weeks 0 and 4 are shown. mRNA-1215 was administered as a prime-boost regimen at weeks 0 and 4 (black arrows). Horizontal dotted lines are set to 0% for (**a**,**b**) and to the neutralization assay LOD for (**c**,**d**). Exact *P* values for all between-group comparisons are presented in Supplementary Table 4.

and G antigens. A measurable CD8⁺ T cell response was detected only against the Pre-F antigen for doses above 10 µg. Notably, mRNA-1215 induced cross-reactive immune responses against the closely related NiV(B) strain, which is responsible for near annual recurrent outbreaks in humans, and against HeV, a related henipavirus, expanding the potential application of this vaccine to outbreaks caused by these viruses.

Overall, mRNA-1215 had a similar safety profile to that of other infectious disease mRNA vaccines, such as COVID-19, influenza, RSV, and Zika[26–30]. Consistent with our findings, mild-to-moderate pain at the injection site, swelling, and redness have been reported in all mRNA vaccine studies. Systemic AEs, such as myalgia, fatigue and headache primarily of mild-to-moderate severity, occurred in the first several days after vaccination, resolving without long-term effects. Two severe AEs, injection site redness and an elevated temperature, occurred. Similar events have been reported in COVID-19 mRNA vaccine recipients. One participant in this trial developed systemic urticaria after the first 100-µg dose of mRNA-1215 that was exacerbated by the second dose. Delayed-onset urticaria has been reported after first and second doses of the COVID-19 mRNA vaccine; it responded to treatment with antihistamines, as occurred with the participant in this study[31,32]. In summary, the safety and tolerability profile of mRNA-1215 was consistent with that of other mRNA vaccines currently undergoing evaluation in clinical studies and with those that have been licensed in many countries.

Most NiV vaccine platforms currently in development focus on the attachment G protein as the immunogen due to its critical function in the initial step of viral entry. Several other platforms have included the fusion (F) antigen, but these vaccines are in the preclinical testing stage[16–18,33]. To the best of our knowledge at the time of this publication, mRNA-1215 is the only vaccine that includes both F and G antigens and that has been tested in a human clinical trial. Including both antigens

in the vaccine has several advantages. First, the amino acid sequence of F is more conserved across the *Henipavirus* genus compared to G, consequently increasing the probability of cross-reactive responses to emerging, related viruses[25]. Second, mRNA-1215 contains a pre-fusion stabilized conformation of the F protein which induces higher neutralizing responses against NiV compared to its post-fusion form[21,22]. Third, both F and G proteins elicit robust T cell responses, including CD8⁺ T cells responses, mainly driven by the F protein[22]. Fourth, in preclinical models, vaccination with either protein protects against NiV challenge[16–18]. Consistent with these findings, preliminary results from nonhuman primate studies showed that vaccination with Pre-F/G mRNA induced humoral and cellular immune responses, which conferred complete protection against NiV challenge[34]. Inclusion of both proteins in the mRNA-1215 vaccine offers a potential advantage in limiting or preventing viral escape that may arise due to mutations driven by selective pressure on one of the proteins. Correlates of protection against NiV infection and disease in humans are yet to be established, underscoring the importance of a vaccine that can induce humoral and cellular immune responses, both of which were induced by the mRNA-1215 vaccine in this trial.

In this study, robust binding and neutralizing antibody induced by mRNA-1215 were detected as early as 2 weeks after the first dose. In comparison, a phase 1, randomized, placebo-controlled trial of the subunit protein vaccine, HeV-sG-V, demonstrated that a single dose provided a limited neutralizing immune response and that two doses were required to elicit a higher antibody response[35]. The neutralizing antibody titers induced by mRNA-1215 remained elevated above baseline out to 1 year after the booster dose. The rapid, robust, and durable immune responses elicited by mRNA-1215 would be advantageous when responding to an acute outbreak or when vaccinating a population

prophylactically for an upcoming seasonal outbreak[5,36], potentially providing immune protection for an extended period of time.

The responses to mRNA-1215 were comparable across dose groups after the prime and up to 5 months after the boost; beyond this time point, the 10-µg group displayed lower antibody titers than the higher-dose groups, yet titers remained above baseline through 1 year of follow-up. These data highlight that the mRNA-1215 vaccine is immunogenic at all dose levels with durable responses for at least 1 year after the boost. Based on these results, doses greater than 10 µg should be evaluated in future clinical trials in at-risk populations where the virus is endemic.

Since 2004, Bangladesh has experienced NiV outbreaks almost every year, resulting in the highest cumulative number of reported NiV cases worldwide[36,37]. There is also concern for future outbreaks of HeV in Australia. Therefore, demonstrating that mRNA-1215 can elicit cross-reactive immune responses against NiV(B) and HeV were important exploratory analyses in this trial. Given the high percentage of amino acid identity between NiV(M) and NiV(B) glycoproteins, mRNA-1215 did in fact elicit a high frequency of cross-reactive memory B cells that recognize both NiV(M) and NiV(B) F and G proteins. There was a low frequency of memory B cells that only bound to the NiV(M) G protein, possibly due to the less conserved amino acid sequence between the G proteins of these two NiV strains. Although the HeV Australia 1994 outbreak strain is more distantly related to NiV(M) (amino acid identity percentage of 89.4% for F and 79.6% for G), NiV(M) antigens in the vaccine elicited cross-reactive HeV binding and neutralizing responses that remained elevated above baseline up to 1 year after vaccination. A proof-of-concept study in nonhuman primates demonstrated that vaccination with mRNA-1215 induced cross-reactive responses to both NiV(B) and HeV, and conferred complete protection following challenge with NiV(B) virus, demonstrating that cross-protection is achievable with this vaccine[34]. These data support potential use of mRNA-1215 against other NiV strains and against closely related henipaviruses, which are currently endemic in Bangladesh and Australia, respectively.

Given the sporadic nature of NiV outbreaks and the low number of individuals infected, it is unlikely that an efficacy study will be possible[38]. Licensure of a NiV vaccine may require acceptance by regulatory authorities that clinical benefit can be demonstrated using a surrogate end point that is considered predictive of efficacy, followed by post-marketing studies that verify vaccine effectiveness. One potential approach could leverage well-controlled animal models of henipavirus infection to demonstrate protective efficacy and identify immune correlates of protection in animal challenge studies[39]. This approach was recently used successfully for the licensure of vaccines to prevent chikungunya disease[40,41]. Considering what could be a surrogate end point for prevention of Nipah virus disease, several studies in nonhuman primates have demonstrated that NiV-specific binding and neutralizing antibody responses are protective after NiV challenge[42,43]; however, the mechanism of protection after vaccination with mRNA-1215 in a NiV challenge model remains to be identified.

There are several limitations to this study. One of these is the small number of participants in each cohort; therefore, the results reported from this trial should be interpreted as descriptive and hypothesis generating. Additionally, the trial was open-labeled and nonrandomized, which could potentially introduce an element of bias in the reporting of reactogenicity. Despite its smaller size, this study provides the first characterization of the safety and humoral and cellular immunogenicity of mRNA-1215. Future studies involving larger cohorts, including individuals at increased risk in endemic regions, are essential to further characterize the safety and immunogenicity profile of this vaccine. Finally, neutralization testing was performed using a well-characterized pseudo-typed virus neutralization assay (PVNA); however, the use of a surrogate viral system that lacks the full structure, entry pathways, and replication cycle of native virus could lead to differences in neutralization potency compared to live virus neutralization assays (LVNAs).

As a result, PVNAs may not fully capture the true biological activity of antibodies under physiological infection conditions which may be better represented in LVNAs.

The mRNA-1215 vaccine is the first structure-based vaccine developed for a prototype pathogen of the *Henipavirus* genus in the *Paramyxovirus* family. The results from our human trial support the application of the chimeric antigen approach to the development of vaccines for other henipaviruses. In addition, the shorter manufacturing timeline for an mRNA vaccine, within several weeks, supports the future development of mRNA-1215 as a countermeasure for rapid deployment in an outbreak situation. This study has addressed the urgent need to develop vaccines against NiV and related pathogens of pandemic potential.

## Online content

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

Aurélie Ploquin [1,5], Rosemarie D. Mason[1,5], LaSonji A. Holman[1], Myra Happe[1], Alicia T. Widge[1], Laura Novik[1], Ana M. Ortega-Villa[2], Galina V. Yamshchikov[1], Ingelise J. Gordon[1], Abidemi Ola[1], Anita Arthur[1], Pamela J. M. Costner[1], Floreliz Mendoza[1], Jamie Saunders[1], Xioalin Wang[1], William R. Whalen[1], Joanna Utoh[1], Jennifer Cunningham[1], Lorin N. Loftus [1], Ashley Heimann [1], Katia Korzeniwsky[1], Shayne F. Andrew [1], Evan Lamb[1], Shruthi Shyam Sunder[1], Amelia Thompson[1], Mary McDonald[1], Kathryn E. Foulds [1], Nancy J. Sullivan[1], Jessica Bahorich[1], Emily E. Coates[1], Rebecca J. Loomis[1], Barney S. Graham[1], Karin Bok[1], Sunny Himansu [3], Brett Leav[3], Walla Dempsey[4], John H. Beigel [4], Mario Roederer[1], Lesia K. Dropulic [1]✉ & The VRC 322 study team*

[1]Vaccine Research Center, the National Institute of Allergy and Infectious Diseases, National Institutes of Health, Bethesda, MD, USA. [2]Biostatistics Research Branch, Division of Clinical Research, the National Institute of Allergy and Infectious Diseases, National Institutes of Health, Bethesda, MD, USA. [3]Moderna, Inc, Cambridge, MA, USA. [4]Division of Microbiology and Infectious Diseases of the National Institute of Allergy and Infectious Diseases, National Institutes of Health, Bethesda, MD, USA. [5]These authors contributed equally: Aurélie Ploquin, Rosemarie D. Mason. *A full list of authors and their affiliations appears at the end of the paper. ✉e-mail: dropulicl@niaid.nih.gov

**The VRC 322 study team**

Maxwell Norris[1], Preeti Apte[1], Renunda Dyer[1], LaShawn Requilman[1], Justine Jones[1], Larisa Strom[1], Tatiana Beresnev[4], Maryam Keshtkar-Jahromi[4], Caitlyn Dulan[1], Li Ou[1], I-Ting Teng[1] & Tongqing Zhou[1]

## Methods

### Study design

This was a phase 1, open-label, dose-escalation, nonrandomized, first-in-human trial to evaluate the safety, tolerability and immunogenicity of mRNA-1215 in healthy adults (ClinicalTrials.gov NCT05398796). Eligible study participants were healthy adults, based on medical history and physical examination, who were 18–60 years of age. Participants were able and willing to complete the informed consent process, provide proof of their identity and were available for clinic follow-up visits for the duration of the trial (52 weeks after the last product administration). The participants were required to have a body mass index of 18–35 kg m$^{-2}$ within 56 days before enrollment.

**Inclusion criteria.** Laboratory-based inclusion criteria within 56 days before enrollment included white blood cell and differential count within the institutional normal range or accompanied by principal investigator (PI) or designee approval; total lymphocyte count ≥800 cells per μl; platelets count of 125,000–500,000 cells per μl; hemoglobin within the institutional normal range or accompanied by PI or designee approval; alanine aminotransferase ≤1.25 × the institutional upper limit of normal (ULN); aspartate aminotransferase ≤1.25 × the institutional ULN; alkaline phosphatase <1.1 × the institutional ULN; total bilirubin within the institutional normal range or accompanied by PI or designee approval; serum creatinine ≤1.1 × the institutional ULN; negative test for HIV infection determined by a US Food and Drug Administration (FDA)-approved method of detection. Inclusion criteria applicable to women of childbearing potential were a negative β-human chorionic gonadotropin pregnancy test (urine or serum) on the day of enrollment, before the first vaccine dose administration, and an agreement to use an effective method of birth control from at least 21 days before enrollment (from receipt of first vaccine dose) through the end of the study.

**Exclusion criteria.** Study participants were excluded from the study if the following conditions applied: females who were breast-feeding or planning to become pregnant during the study; treatment with systemic immunosuppressive medications for more than 10 days or with cytotoxic medications within the 4 weeks before enrollment or any treatment within the 14 days before enrollment; receipt of blood products within 16 weeks before enrollment; receipt of any vaccine, including COVID-19 vaccines, within 4 weeks before enrollment; receipt of an investigational research agent(s) within 4 weeks before enrollment or planned receipt of investigational products while on the study; current allergy treatment with allergen immunotherapy with antigen injections, unless the participant was on a maintenance schedule; current anti-TB prophylaxis or therapy; a known immediate hypersensitivity to any component of the study product, including polyethylene glycol; a confirmed past NiV infection, previous residence in (for more than 6 months), or planned travel for any length of time during the study to countries where NiV infection is endemic, such as Bangladesh, India or the Philippines. In addition, participants with a history of any of the following medical conditions were excluded: serious reactions to vaccines, including allergic reactions, such as anaphylaxis, urticaria or other allergic reaction requiring medical intervention, to SARS-CoV-2 mRNA vaccines, as determined by the investigator; a history of myocarditis and/or pericarditis, hereditary angioedema, acquired angioedema, or idiopathic forms of angioedema; asthma that is not well controlled; diabetes mellitus (type I or II), with the exception of gestational diabetes; thyroid disease that is not well controlled; idiopathic urticaria within the past year; autoimmune disease or immunodeficiency; hypertension that is not well controlled; bleeding disorder diagnosed by a physician (for example factor deficiency, coagulopathy, or platelet disorder requiring special precautions) or more-than-expected bruising or bleeding difficulties with intramuscular (i.m.) injections or blood draws; malignancy that is active or a history

of malignancy that is likely to recur during the period of the study; a seizure disorder other than: (1) febrile seizures; (2) seizures secondary to alcohol withdrawal more than 3 years ago; or (3) seizures that have not required treatment within the last 3 years; asplenia, functional asplenia or any condition resulting in the absence or removal of the spleen; Guillain–Barré syndrome; any medical, psychiatric or social condition, occupational reason or other responsibility that, in the judgment of the investigator, is a contraindication to protocol participation or that impairs a participant's ability to give informed consent, including but not limited to certain clinical presentations of infectious diseases, drug or alcohol abuse, autoimmune diseases, psychiatric disorders, or heart disease. Because the effects of the vaccine on the fetus are not known, pregnant women were not eligible for the trial. Children were not eligible to participate in this clinical trial because the investigational vaccine had not been previously evaluated in adults.

The study was sponsored by the National Institute of Allergy and Infectious Diseases (NIAID) Division of Microbiology and Infectious Diseases and conducted at the National Institutes of Health (NIH) Clinical Center by the Vaccine Research Center Clinical Trials Program, NIAID, NIH, in Bethesda, MD after approval by the NIH Institutional Review Board. Participants were recruited from the greater Washington, DC area using institutional review board-approved recruitment advertisements and community outreach and engagement. Written informed consent was obtained from all participants before enrollment in the study. Participants were compensated for their time and inconvenience associated with participation in the trial.

### Trial vaccine

The investigational mRNA-1215 vaccine is a lipid nanoparticle (LNP) dispersion containing mRNA that encodes for a secreted pre-fusion stabilized F component covalently linked to a G monomer (Pre-F/G of a NiV Malaysian 1999 strain (isolate NV/MY/99/VRI-2794), GenBank AJ564621) with a trimerization domain resulting in secretion of a trimer of heterodimers[21,22]. The LNPs are composed of four lipids: one proprietary ionizable lipid, SM-102, and three commercially available lipids including cholesterol, DSPC (dioleoyl-glycero-phosphocholine) and DMG-PEG2000 (di-myristoyl diglyceride-polyethylene glycol). The ratio of mRNA to lipid is constant across all doses used in this trial. mRNA-1215 is formulated in 20 mM Tris buffer, 87 mg ml$^{-1}$ sucrose and 10.1 mM acetate at a pH of 7.5. The vaccine was co-developed by the VRC, NIAID and Moderna, Inc. and manufactured by Moderna, Inc.

### Study procedures

The trial evaluated a dose escalation of the mRNA-1215 vaccine from 25 μg to 50 μg to 100 μg. Each dose was administered intramuscularly (i.m.) in the deltoid muscle by needle and syringe as a two-dose regimen on day 0 and day 28 to 10 participants per dosing group. Enrollment began in the 25-μg dose group. After the protocol safety review team (PSRT) assessed the safety data at 2 weeks after the first vaccination for the first three 25-μg dose recipients and approved dose escalation, the enrollment began for the 50-μg dose group, while sequential enrollment continued in the 25-μg dose group. A similar dose-escalation approval occurred to open the 100-μg dose group for enrollment. After three participants received the 100-μg dose, the PSRT determined that it was safe to continue dosing in the 100-μg group and enrollment into the 50-μg and 100-μg groups alternated in a sequential fashion. After enrollment of the 25-, 50- and 100-μg dose groups was completed, a protocol-specified interim analysis of the cumulative safety and immunogenicity data from 2 weeks after the second vaccine administration was conducted, and a fourth dose, 10 μg, was selected and administered as a two-dose regimen 28 days apart to an additional group of ten participants.

Study participants self-reported sex (male or female), race (American Indian/Alaska Native, Asian, Native Hawaiian or other Pacific Islander, Black or African American, White, Multiracial

or unknown/not reported), and ethnic group (Hispanic or Latin American or Not Hispanic or Latin American background) on a demographic form provided to them after enrollment. Subsequently, these data were entered into the study database. Study investigators conducted clinical assessments of participants pre-vaccination and at scheduled visits throughout the study with the last study visit occurring 52 weeks after the second vaccination. Participants were observed for at least 30 min after vaccination, and at the end of the observation period, vital signs were obtained, and the injection site was assessed. A phone evaluation was conducted the day after each vaccination, and a clinic visit was established, if indicated, based on symptoms and signs reported by the participant. Blood for clinical laboratory testing to assess safety was obtained before each vaccination and at various time points after vaccination and included a complete blood count with differential, total bilirubin, aspartate and alanine aminotransferase (AST and ALT), alkaline phosphatase and creatinine. A pregnancy test was obtained before each vaccination for women of reproductive potential and was confirmed to be negative before vaccine administration. Blood was also collected for evaluation of immunogenicity using serum, plasma and peripheral blood mononuclear cells (PBMCs).

Participants recorded solicited local and systemic AEs (reactogenicity signs and symptoms) and concomitant medications on a diary card, daily, for 7 days after each vaccination. Unsolicited AEs were collected for 28 days after each vaccine administration. The severity of AEs was determined using the Toxicity Grading Scale for Healthy Adults and Adolescent Volunteers Enrolled in Preventive Vaccine Clinical Trials (US FDA Administration Guidance-September 2007). Serious AEs (SAEs), AEs leading to withdrawal from the study, new-onset chronic medical conditions and AEs of special interest were collected for assessment of safety after the first vaccination through the last study visit. AEs of special interest included thrombocytopenia, new-onset or worsening neurological diseases (Guillain–Barré syndrome, acute disseminated encephalomyelitis, idiopathic peripheral facial nerve palsy or seizures), anaphylaxis, myocarditis, pericarditis and myopericarditis. Medically attended AEs were collected through 6 months after the second dose.

All secondary and exploratory immunogenicity assessments in this trial were performed using the same cohort of 40 participants: 18 females and 22 males with a mean age of 37 years (range 22–59); therefore, sex and age of the participants were consistent across all assessments.

The clinical trial protocol was amended several times during the course of trial. One amendment, implemented on 20 July 2022, was made in response to a US FDA request to add additional information on product preparation to the protocol (section 7.3.1). Another amendment, implemented on 15 June 2023, updated the group 4 dose (10 μg) and its schedule of administration from a 12-week interval to a 4-week interval and added safety information for groups 1–3 (25, 50 and 100 μg). Additional amendments were related to the change of the study PI and were implemented on 22 December 2022, 20 December 2023 and 15 June 2023.

There were a total of 16 minor protocol deviations that included: ten follow-up visits that occurred out-of-window (ranging from 1–8 days); one missed visit due to participant's illness; one incomplete sample collection due to side effects experienced by the study participant during an apheresis procedure; two incomplete blood draws due to inability to collect more than 100 ml of the required 120 ml of blood; one incident of a blood collection of 6 ml over the allowable 550 ml per 8-week limit; and one incident of a participant dating the informed consent with the incorrect year. All the protocol deviations were minor and had no impact on the safety of the study participants or integrity of the study results.

## Study objectives

The primary objective of the trial was to evaluate the safety and tolerability of the two-dose vaccination regimen at each dose. The secondary objective was to evaluate Pre-F and G antibody responses to the mRNA-1215 vaccine at 2 weeks after the second vaccination at each dose level. Exploratory objectives included evaluation of Pre-F and G antibody responses, neutralizing antibody responses, and antigen-specific T and B cell responses at various time points throughout the study.

## Statistical analysis

All participants who received at least one vaccination were included in the analysis of safety and reactogenicity and in the immunogenicity analysis. One participant in the 10-μg dose group, received only the first dose of vaccine, and hence, did not have a visit at 2 weeks after the second dose, the secondary immunogenicity time point. One participant in the 25-μg group missed the visit at 2 weeks after the second dose and also did not have the secondary immunogenicity time point. We conducted a power calculation, based on our primary safety objective, to determine our ability to detect SAEs with group sample sizes of ten. For each group size of ten participants, there was over a 90% chance to observe at least one SAE if the true event rate was at least 0.21, and over a 90% chance to observe no SAE if the true rate was no more than 0.01. To compare immune responses between dose groups, comparisons were made using ANOVA, followed by pairwise two-sample $t$-tests after $\log_{10}$ transformation of raw titers. Within-group time point comparisons were performed using paired $t$-tests after $\log_{10}$ transformation of raw titers. Reported GMTs and 95% CIs are asymptotic. Given the small sample size of this trial, group comparisons are reported descriptively. The study was not designed to detect large immunological differences between the dose groups (1.2 × s.d. of the immune response with 80% power to detect between-group differences if the mean difference between groups is at least 1.2 times the standard deviation of the immune response). To maintain statistical power in this smaller study, adjustments for multiple comparisons were not performed. A post hoc analysis of the ELISA immunogenicity analysis was performed by implementing mixed models for repeated measures with effects for week, dose group and interaction between week and dose group using the mmrm package in R[44]. Statistical analyses for the exploratory end points that evaluated NiV(B) neutralization and HeV Pre-F and G ELISA and neutralization assays were conducted in the post hoc manner. No sex-based analyses were performed; because of the small size of our study, such analyses were not preplanned. All data were graphed in Prism (v.10.4.1).

All statistical tests were two-sided. For ELISA binding assays, all titer values, including those below the LLOQ, were detected and used in the analysis. For neutralization assays, titer values that were negative were imputed as the reciprocal of half of the lowest dilution tested, which was 1:50, the LOD. For both ELISA and neutralization assays, measurements were obtained through repeated assessment of the same sample at a single time point, and the mean of the replicates was used for statistical analyses. Analyses were performed using R v.4.3.3, and results were considered statistically significant if $P < 0.05$.

## Assessment of anti-Pre-F and anti-G IgG binding antibody responses

The methods for the NiV IgG ELISA have been previously described[22], and the following modifications were implemented in this study. Immulon 4HBX 384-well ELISA plates (Thermo Scientific) were coated with 40 ng per well of NiV Malaysia 1999 isolate NV/MY/99/VRI-2794 (GenBank AJ564621) stabilized Pre-F or monomeric G protein in 0.2 M BupH carbonate-bicarbonate buffer, pH 9.4 (Thermo Scientific) at 4 °C for 16 h. For HeV assays, 80 ng per well of Pre-F and monomeric G from HeV Australia 1994 (GenBank AF017149) viral isolate was used for plate coating. Each sample was run on two separate 384-well ELISA plates coated with Pre-F or monomeric G protein. After three washes in 1× PBS/0.05% Tween-20 (PBS-T), the plates were blocked with 1× PBS/20%FBS for 1 h at room temperature and then washed again in PBS-T. Serum samples were diluted as fourfold serial dilutions in

1× PBS/1% FBS/0.2% Tween-20 solution, added to the antigen-coated plates, and incubated for 45 min at room temperature. Following PBS-T washes, anti-nonhuman primate IgG-horseradish peroxidase conjugate (Southern Biotech) cross-reacting to human IgG was prepared in 1× PBS/2% milk and added to the plates for a 45-min incubation at room temperature. After final washes with PBS-T, SureBlue 3,5,3′,5′-tetramethylbenzidine (TMB) (SeraCare KPL) was used as the substrate to detect antibody responses. The reactions were stopped with 1 N $H_2SO_4$ sulfuric acid (Sigma Aldrich) within 15 min. Optical density (OD) measurements were obtained after reading the plates at 450 nm with an Envision plate reader (PerkinElmer). The value of the antibody response was determined by obtaining half of the maximum effective concentration ($EC_{50}$) titer from the OD450 nm curves for the function of the reciprocal dilution using a 4-PL curve fit in (Prism, v.10.2.2 software). NiV Pre-F and G $EC_{50}$ titers were normalized using the International Standard and reported as IU ml$^{-1}$, whereas HeV Pre-F and G $EC_{50}$ titers were not normalized and reported as such. LLOQ for NiV assays were determined to be 10.15 IU ml$^{-1}$ for Pre-F and 16.32 IU ml$^{-1}$ for G, respectively. For assessment of anti-Hendra virus binding antibody responses, the Pre-F and G proteins from HeV Australia (1994 AF017149) were used to coat the plates and assays were run as described above.

### Assessment of neutralizing antibody responses

Serum neutralizing activity was measured in a luciferase reporter gene assay using single round infection of HEK293T (ATCC CRL-11268) target cells with pseudo-typed virus particles (pseudovirus). Pseudovirus was produced according to the manufacturer's protocol via Lipofectamine3000 (Thermo Fisher) transfection of HEK293T cells with DNA plasmids encoding NiV full-length F and truncated G (GΔ34)[45] proteins in combination with a luciferase reporter plasmid (VRC5601: pHR′ CMV Luc), and a plasmid containing the essential HIV structural genes (VRC5602: pCMV DR8.2) at a plasmid ratio of 6:6:1:1 F:GΔ34:VRC5601:VRC5602. Neutralization was measured by adding 10 µl of serum (nine threefold dilutions ranging from 1:50 to 1:328,050) in complete Dulbecco's modified Eagle medium (cDMEM), no phenol red, supplemented with 10% heat-inactivated fetal bovine serum (FBS), 0.6 mg ml$^{-1}$ L-glutamine and 1% penicillin–streptomycin (all from Thermo Fisher) and added to 10 µl of pseudovirus (at a concentration selected to yield relative light units (RLU) of at least ten times above background) in black ViewPlate 384-F tissue culture-treated plates (Perkin Elmer) with a white adhesive bottom seal (Revvity) that were incubated at 37 °C with 5% $CO_2$, 95% humidity. After 30 min, 20 µl of HEK293T cells (~2,600 cells per well) were added to all wells and incubated at 37 °C with 5% $CO_2$, 95% humidity. Each experiment plate contained a column of cells only (no serum or virus) as a control for background luciferase activity and a column of virus and cells only (no serum) as a maximal viral entry control. After 72 h, 40 µl Bright-Glo luciferase assay substrate (Promega) was added to each well, mixed by pipetting, and RLU was measured at 570 nm on a SpectraMax L luminometer (Molecular Devices). The 50% inhibitory dilution ($ID_{50}$), defined as the serum dilution required to achieve 50% reduction in RLU compared to virus control wells after subtraction of background RLUs, was obtained using an asymmetric five parameter fit curve (Prism v.10.0.0). For neutralization curves that did not reach 20% neutralization (background), $ID_{50}$ is reported as 25 (below the lowest (50-fold) dilution tested), whereas for neutralization curves that reached below 50% but above 20% neutralization, $ID_{50}$ was reported as 50 (the lowest (50-fold) dilution tested). $ID_{50}$ values are derived from three NiV Malaysia (GenBank AJ564621) and HeV Redlands (GenBank HM044317) assays or two NiV Bangladesh 2004 (isolate 810398) Hu - MK673564 isolate) assays with two technical replicates per assay. NiV(M) and NiV(B) $ID_{50}$ titers were normalized using the International Standard and reported as IU ml$^{-1}$, whereas HeV $ID_{50}$ titers were not normalized and reported as such. The LOD for neutralization assays was determined to be 17 IU ml$^{-1}$ for NiV(M), 9 IU ml$^{-1}$ for NiV(B) and an $ID_{50}$ titer of 50 for HeV.

### Normalization of NiV immunogenicity data to the WHO/NIBSC NiV international standard

All NiV virus ELISA binding and neutralization assays included the WHO International Standard for anti-NiV antibodies as an additional sample (National Institute for Biological Standards and Control (NIBSC) code 22/130) to enable harmonization of the ELISA binding and neutralization data to a common unit, thereby facilitating comparison to different NiV vaccine studies and assays[46]. The International Standard is based on a pool of sera from 36 NiV convalescent individuals: 6 from Malaysia and 30 from Bangladesh. Normalization of $ED_{50}$ or $IC_{50}$ titers was achieved by using the equation: $EC_{50}$ or $ID_{50}$ of sample × 1,000/ $EC_{50}$ or $ID_{50}$ International Standard median across assays. Thus, the International Standard had an attributed value of 1,000 IU ml$^{-1}$ for each graphical representation of data of anti-NiV antibody responses. An International Standard for F glycoprotein binding was not established due to the lack of enough data for anti-Pre-F assays during the International Standard characterization study[5]. However, we decided to use the International Standard for the normalization of anti-NiV Pre-F antibody titers. Due to the lack of an International Standard for HeV, HeV binding and neutralizing titers are expressed as $EC_{50}$ and $ID_{50}$ titers, respectively.

### Assessment of B cell responses by B cell probe binding

Cryopreserved PBMCs were thawed, washed briefly with phenol-free RPMI/4% heat-inactivated newborn calf serum (R&D Systems), and incubated with an aqua live/dead fixable dead cell stain kit (Thermo Fisher Scientific) for 20 min at room temperature. Cells were stained with the following antibodies (all monoclonal unless indicated) for 20 min at room temperature: IgD FITC (1:160 dilution) (goat polyclonal, Southern Biotech), IgM PerCP-Cy5.5 (1:20 dilution) (clone G20-127, BD Biosciences), IgA Dylight 405 (1:80 dilution) (goat polyclonal, Jackson Immunoresearch), CD20 BV570 (1:40 dilution) (clone 2H7, BioLegend), CD27 BV650 (1:20 dilution) (clone O323, BioLegend), CD14 BV785 (1:80 dilution) (clone M5E2, BioLegend), CD16 BUV496 (1:80 dilution) (clone 3G8, BD Biosciences), CD4 BUV737 (1:320 dilution) (clone SK3, BD Biosciences), CD19 APC (1:80 dilution) (clone J3-119, Beckman), IgG Alexa 700 (1:20 dilution) (clone G18-145, BD Biosciences), CD3 APC-Cy7 (1:40 dilution) (clone SP34-2, BD Biosciences), CD38 PE (1:160 dilution) (clone OKT10, Caprico Biotechnologies), CD21 PE-Cy5 (1:40 dilution) (clone B-ly4, BD Biosciences) and CXCR5 PE-Cy7 (1:40 dilution) (clone MU5UBEE, Thermo Fisher Scientific). Stained cells were then incubated with matched Pre-F trimer and G trimer probe pairs: streptavidin-BUV661 (BD Biosciences) labeled Bangladesh Pre-F probe (1:25 dilution) and streptavidin-BUV395 labeled Malaysia Pre-F probe (1:25 dilution), and streptavidin-BUV661 (BD Biosciences) labeled Bangladesh G probe (1:25 dilution) and streptavidin-BUV395 labeled Malaysia G probe (1:25 dilution) for 30 min at 4 °C (protected from light). The Pre-F probe used in these assays (GenBank AJ627196) is derived from a NiV Malaysia 1999 isolate that differs by only one amino acid, at position T348M, from AJ564621, the isolate sequence used for mRNA-1215. The NiV Malaysian G probe matched the sequence of G in the vaccine. For the Bangladesh probes, both the Pre-F and G amino acid sequences were derived from the 2004 sequence Genebank MK673564. Cells were washed and fixed in 0.5% formaldehyde (Tousimis Research Corp) before data acquisition. All antibodies were titrated on human PBMCs to determine the optimal concentration. Samples were acquired on a BD FACSymphony cytometer and analyzed for the B cell phenotype using FlowJo v.10.10.0 (BD).

### Assessment of T cell responses by intracellular cytokine staining

Cryopreserved PBMCs were thawed and incubated overnight at 37 °C/5% $CO_2$. After incubation, cells were stimulated with Malaysia Pre-F or Malaysia G peptide pools (Malaysia 1999 isolate NV/MY/99/ VRI-2794 (GenBank AJ564621); JPT Peptides) at a final concentration

of 2 µg ml⁻¹ in the presence of 3 mM monensin for 6 h. The Pre-F and G peptide pools consist of 123 and 108 individual peptides, respectively, as 15mers overlapping by 11 amino acids in 100% dimethylsulfoxide (DMSO). Negative controls received an equal concentration of DMSO to that of peptide pools (final concentration of 0.5%). Intracellular cytokine staining was performed as described previously[47]. The following monoclonal antibodies were used: CD3 APC-Cy7 (1:160 dilution) (clone SP34.2, BD Biosciences), CD4 PE-Cy5.5 (1:80 dilution) (clone SK3, Thermo Fisher), CD8 BV570 (1:40 dilution) (clone RPA-T8, BioLegend), CD45RA PE-Cy5 (1:2,500 dilution) (clone 5H9, BD Biosciences), CCR7 BV650 (1:10 dilution) (clone G043H7, BioLegend), CXCR5 PE (1:10 dilution) (clone MU5UBEE, Thermo Fisher), CXCR3 BV711 (1:20 dilution) (clone 1C6/CXCR3, BD Biosciences), PD-1 BUV737 (1:20 dilution) (clone EH12.1, BD Biosciences), ICOS Pe-Cy7 (1:80 dilution) (clone C398.4A, BioLegend), CD69 ECD (1:40 dilution) (cloneTP1.55.3, Beckman Coulter), IFNγ Ax700 (1:320 dilution) (clone B27, BioLegend), IL-2 BV750 (1:40 dilution) (clone MQ1-17H12, BD Biosciences), IL-4 BB700 (1:20 dilution) (clone MP4-25D2, BD Biosciences), TNF-FITC (1:80 dilution) (clone Mab11, BD Biosciences), IL-13 BV421 (1:20 dilution) (clone JES10-5A2, BD Biosciences), IL-17 BV605 (1:20 dilution) (clone BL168, BioLegend), IL-21 Ax647 (1:10 dilution) (clone 3A3-N2.1, BD Biosciences) and CD154 BV785 (1:20 dilution) (clone 24-31, BioLegend). Aqua live/dead fixable dead cell stain kit (Thermo Fisher Scientific) was used to exclude dead cells. All antibodies were previously titrated to determine the optimal concentration. Samples were acquired on a BD FACSymphony flow cytometer and analyzed using FlowJo v.10.10.0 (TreeStar).

## Reporting summary

Further information on research design is available in the Nature Portfolio Reporting Summary linked to this article.

## Data availability

Results generated in this study are available as de-identified data on ClinicalTrials.gov under identifier NCT05398796. The study protocol and informed consent form are available on ClinicalTrials.gov under identifier NCT05398796. Additional data, including de-identified individual level data, may be made available upon reasonable request to the corresponding author, sent to dropulicl@niaid.nih.gov, for investigators whose proposed use of the data has been approved by the NIH Institutional Review Board. The anticipated timeframe for a response to such a request is about 1 to 3 weeks. A signed data access agreement is required before data sharing. Access to data is dependent on the time to establish a signed agreement between the NIH and the requesting investigator. This may take up to 1 to 3 months. Access to data is usually not time limited after a signed agreement is executed and data are transferred. Source data are provided with this paper.

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

## Acknowledgements

This research was supported (in part) by the Intramural Research Program of the NIH (project/award no. ZIAAI005047). The contributions of the NIH author(s) were made as part of their official duties as NIH federal employees, are in compliance with agency policy requirements, and are considered Works of the US Government; however, the findings and conclusions presented in this paper are those of the author(s) and do not necessarily reflect the views of the NIH or the US Department of Health and Human Services (A.P., R.D.M., A.T.W., L.H., L.N., M.H., A.M.O.-V., G.V.Y., E.C., I.J.G., A.O., A.A., P.J.M.C., F.M., J.S., X.W., W.W., J.U., J.C., L.N.L., A.H., K.K., S.F.A., E.L., S.S.S., A.T., M.M., K.E.F., N.J.S., J.B., R.J.L., B.S.G., K.B., M.R. and L.K.D). Moderna manufactured the vaccine. The investigators of the NIAID Vaccine Research Center and the Division of Microbiology and Infectious Diseases (W.D. and J.H.B.) who sponsored the study, had oversight over study conceptualization and design, data collection and analysis, and made the decision to publish the study results and prepare this manuscript. We thank the trial participants for their time and dedication to this study. We also thank other NIAID staff for their support and contributions to enable this study: P. Williams, K. Brooks, A. Kwiecien, R. S. Rothwell, S. Plummer, A. Eshun, S. Trammel, J. Phillips, S. Yuan, L. Wu, M. Nason, R. Du and J. Moliva. We thank the staff of The EMMES Company (E. Burch) and of Technical Resources International for supporting data management activities and regulatory submissions for the protocol. We thank the first WHO collaborative study for establishing the anti-Nipah virus antibody international standard (WHO/BS/2023.2458), coordinated by the NIBSC, part of the Medicines and Healthcare Products Regulatory Agency, in partnership with the Coalition for Epidemic Preparedness Innovations (CEPI), the Universiti Malaya, Malaysia, and the International Center for Diarrhoeal Disease Research, Bangladesh (icddr,b), for allowing us to use the anti-NiV antibody international standard to normalize our binding antibody (22/130 bd) and neutralization (22/130 nt) assay data in IU ml⁻¹.

## Author contributions

Study concept and design: A.T.W., L.N., L.A.H., I.J.G., A.O., G.V.Y., E.E.C., M.R., A.P, L.N.L., S.S.S., R.M., N.J.S., R.J.L., K.E.F., B.L., B.S.G., K.B., S.H., W.D., J.H.B. and L.K.D; Data collection: A.T.W., L.A.H., L.N., I.J.G., A.O., A.A., P.J.M.C., F.M., J.S., X.W., W.W., J.U., J.C., A.A., A.P., R.D.M., L.N.L., S.S.S., A.H., K.K., A.T., M.M., K.E.F., S.F.A. and E.L.; Data analysis: L.N., L.A.H., G.V.Y., M.H., E.E.C., M.R., A.P., L.L., S.S.S., R.D.M., A.H., K.K., A.M.O.-V., K.E.F., S.F.A., E.L. and L.K.D.; Data interpretation: L.N., L.A.H., G.V.Y., M.H., E.E.C., A.P., R.D.M., A.M.O.-V., K.E.F., M.R. and L.K.D.; Drafting and revisions: L.N., L.A.H., G.V.Y., M.H., E.E.C., M.R., A.P., R.D.M., A.M.O.-V., K.E.F. and L.K.D.; and Review and comments: all co-authors.

## Competing interests

R.J.L. and B.S.G. are named as inventors on a US patent 11890339 entitled 'Nipah Virus Immunogens and Their Use', which is a provisional patent filed by the Department of Health and Human Services. B.L. and S. H. are employees of and own shares of Moderna. B.S.G. declares a current affiliation with Morehouse School of Medicine, Atlanta. A.P., R.D.M., L.A.H., M.H., A.T.W., L.N., A.M.O.-V., G.V.Y., I.J.G., A.O., A.A., P.J.M.C., F.M., J.M., X.W., W.R.W., J.U., J.C., L.N.L., A.H., K.K., S.F.A., E.L., S.S.S., A.T., M.M., K.E.F., N.J.S., J.B., E.E.C., K.B., W.D., J.H.B., M.R. and L.K.D. declare no competing interests.

## Additional information

**Extended data** is available for this paper at https://doi.org/10.1038/s41591-026-04265-1.

**Correspondence and requests for materials** should be addressed to Lesia K. Dropulic.

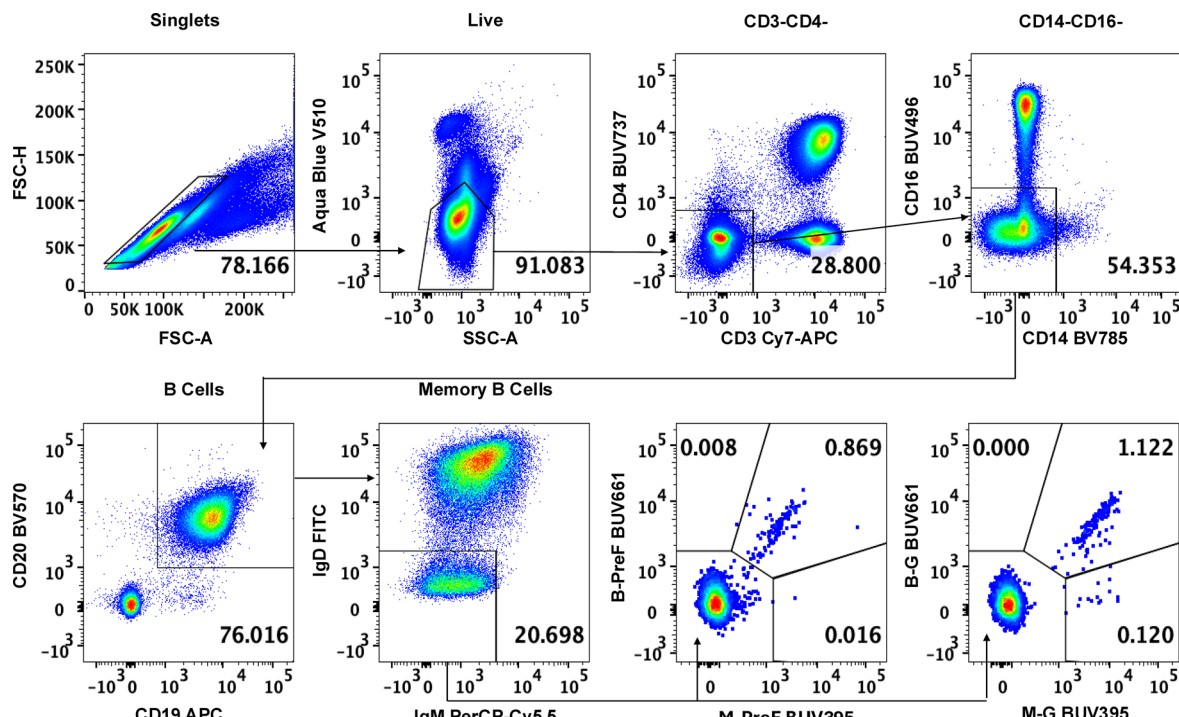

**Extended Data Fig. 1 | B cell gating strategy.** Representative flow cytometry plots displaying the gating strategy for the B cell analysis in Fig. 4. Cells were gated as singlets and live cells using forward-area and height, and side scatter and Aqua Blue. CD3 and CD4 double negative cells were then selected based on the absence of expression of both CD14 and CD16. Memory B cells that were double positive for CD20 and CD19 were then selected based on the lack of IgD and IgM expression. Their binding specificity was determined using Bangladesh (B) and Malaysia (M) Pre-F and G probes.

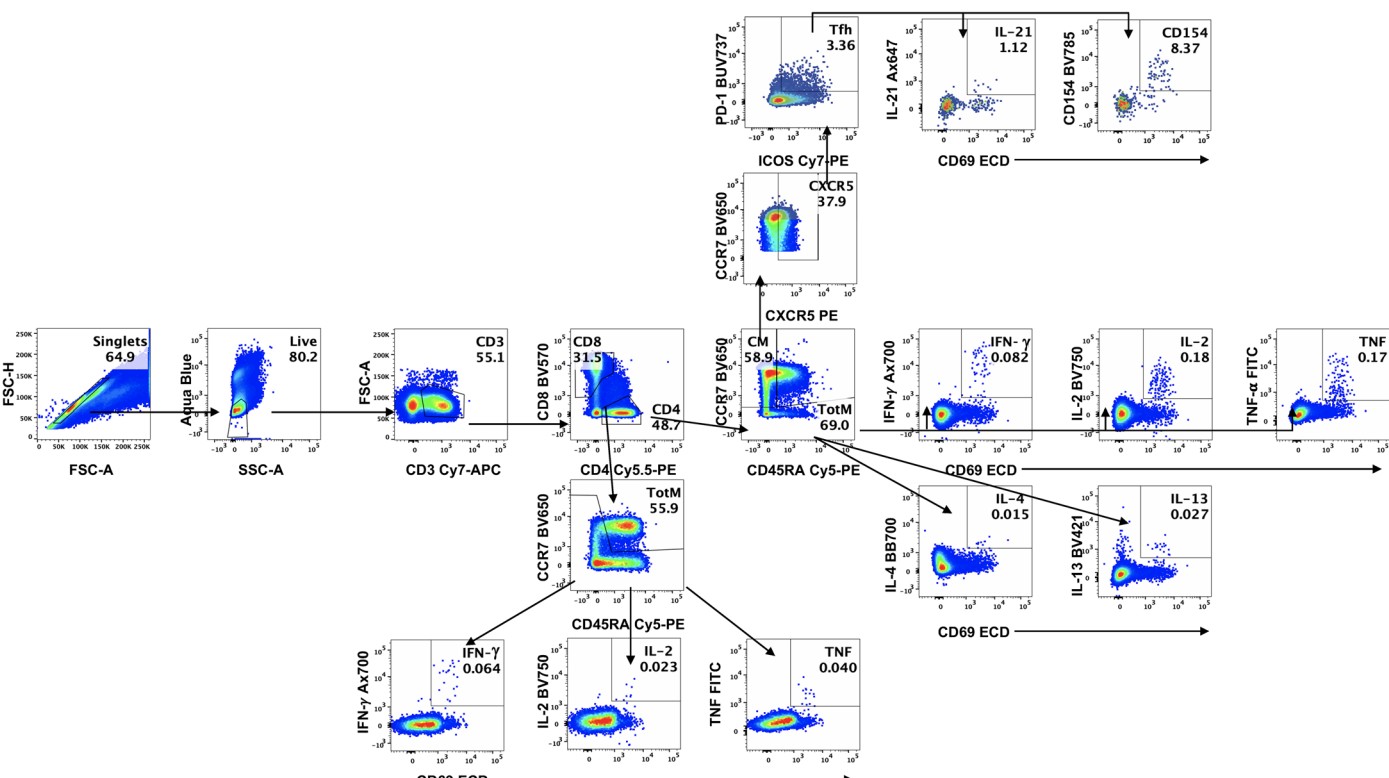

**Extended Data Fig. 2 | Th1 Th2 Tfh Gating.** Representative flow cytometry plots displaying the gating strategy for the T cell analysis in Extended Data Fig. 4. First, single cells were gated, followed by AquaBlue-low (live) cells, then CD3+ T cells. Second, memory subsets were identified from CD4+ and CD8+ T cell populations. Central memory (CM) T cells (CCR7+CD45RA−) and total memory (TotM) T cells composed of effector memory (CCR7−CD45RA−) and terminal memory cells (CCR7−CD45RA+) cells were gated, while excluding naive cells (CCR7+CD45RA+). Third, T follicular helper (T_fh) cells were identified by selecting CXCR5+ CM T cells, followed by PD-1+ and ICOS+ populations. Lastly, CD69+, cytokine+ gates were drawn to identify cytokine-expressing populations in memory CD4 T cells, memory CD8 T cells, and T_fh cells.

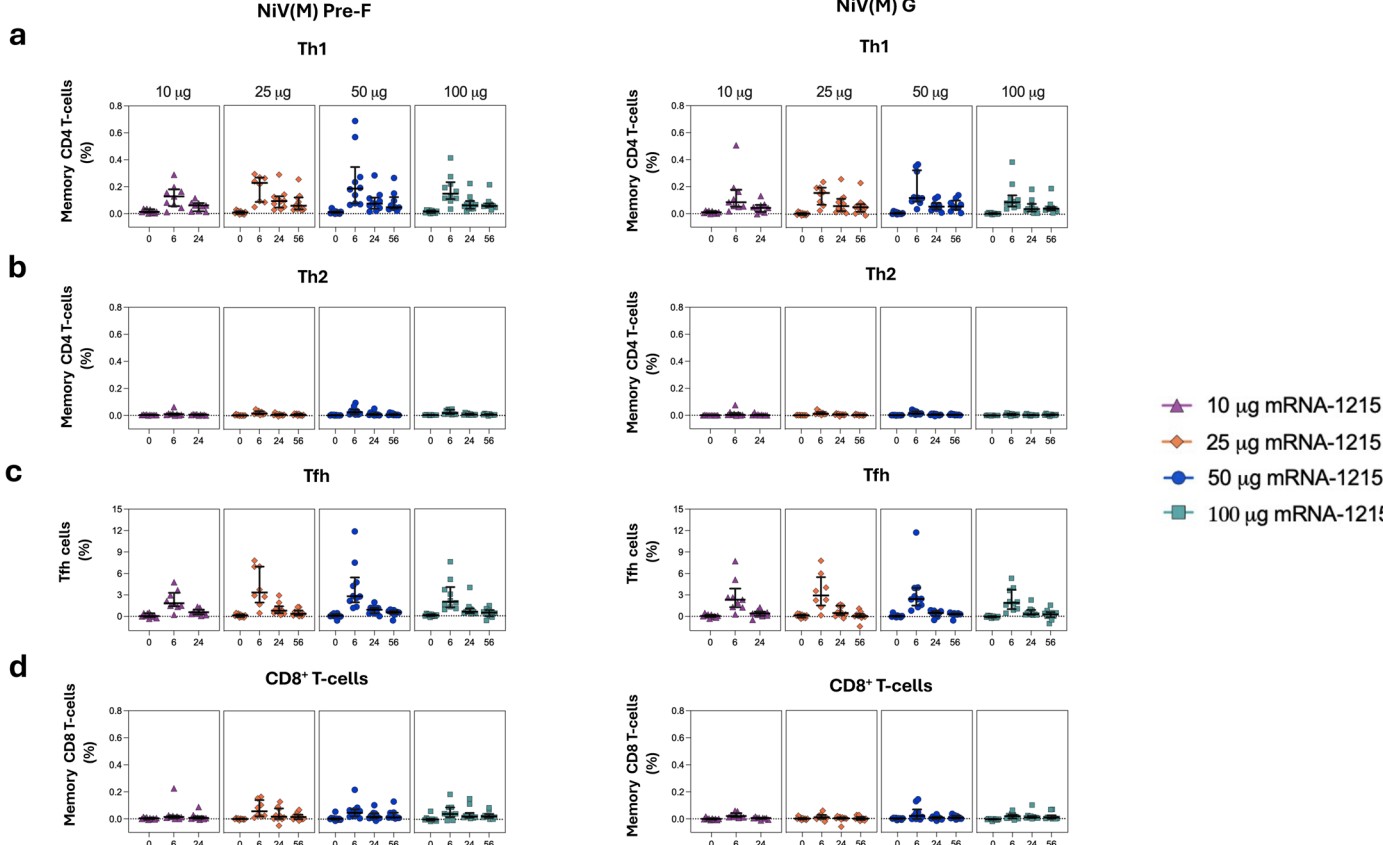

**Extended Data Fig. 3 | Antigen-specific NiV(M) T cell responses.** T cell responses to Pre-F and G peptide pools were assessed by intracellular cytokine staining using peripheral blood mononuclear cells. **a**, $T_{h1}$ responses (interleukin (IL)-2, interferon (IFN)-$\gamma$ or tumor necrosis factor (TNF-a); **b**, $T_{h2}$ responses (IL-4 or IL-13), **c**, $T_{fh}$ responses (CD40L or IL-21), **d**, CD8+ T cell responses. Purple triangles, orange diamonds, blue circles, and teal squares represent individual participants who received 10, 25, 50, or 100 μg doses of mRNA-1215 vaccine,

respectively. Group geometric mean titers (GMTs) are indicated by horizontal lines and 95% confidence intervals (CIs) as error bars. Results in **a-d** are shown for $n = 10$ participants in each dose group; one participant in the 10 μg group received only a single dose of mRNA-1215 and only baseline (Week 0) are shown for this participant. mRNA-1215 was administered as a prime-boost regimen at weeks 0 and 4. Horizontal dotted lines are set to 0%.

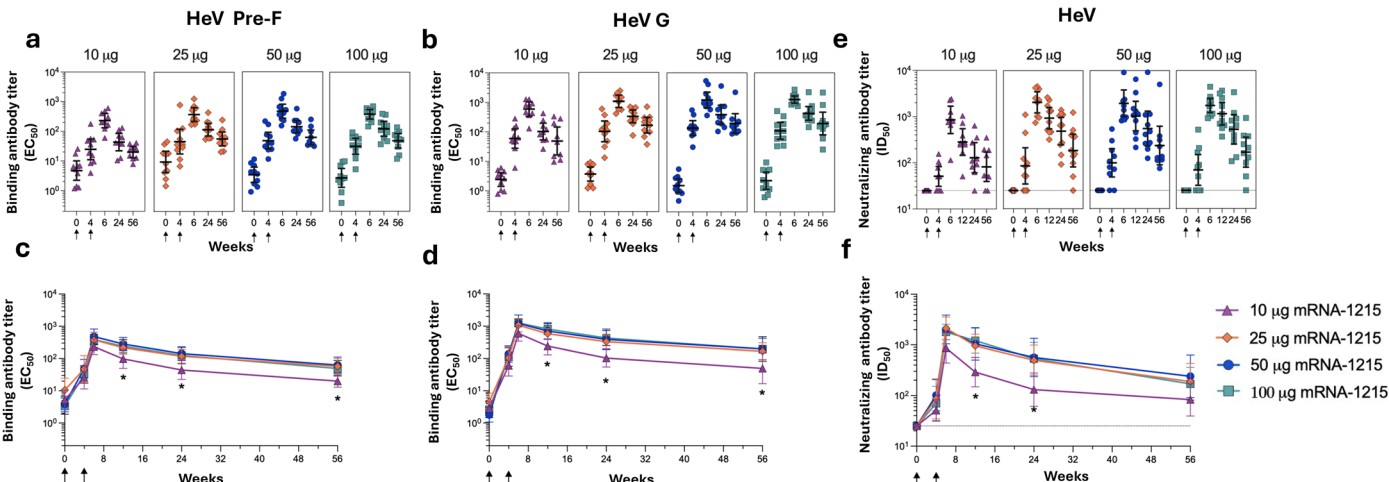

**Extended Data Fig. 4 | HeV cross-reactive vaccine-induced binding and neutralizing antibody titers.** HeV-specific binding antibody titers against Pre-F (**a, c**) and G (**b, d**) were measured by ELISA and HeV neutralizing antibody titers (**e, f**) were assessed by pseudovirus neutralization assay. Panels **a, b**, and **e**, show individual participant titers (geometric shapes) at selected time points with group geometric mean titers (GMTs) indicated by horizontal lines and 95% confidence intervals (CIs) as error bars. Panels **c, d**, and **f** display GMTs over time (geometric shapes) with 95% CIs (error bars). Geometric shapes, purple triangles, orange diamonds, blue circles, and teal squares, represent the 10, 25, 50, and 100 μg dose groups, respectively. Binding and neutralizing titers are expressed as a reciprocal $EC_{50}$ (half maximal effective concentration) titers or reciprocal $ID_{50}$ (50% inhibitory dose). Asterisks represent statistically significant differences between the GMTs of the 10 μg and the 25, 50, or 100 μg dose groups at week 12 (binding antibody titer: Pre-F p-value (p) ≤ 0.030; G p ≤ 0.002; neutralizing antibody titer p ≤ 0.001); week 24 (binding antibody titer: Pre-F p ≤ 0.008, G p ≤ 0.005; neutralizing antibody titer p ≤ 0.010), and week 56 (binding antibody titer: Pre-F p ≤ 0.011; G p ≤ 0.025)^, determined by ANOVA and uncorrected pairwise two-sided t-test; adjustments for multiple comparisons were not performed. Results in **a-f** are shown for $n = 10$ participants in each dose group except for one participant in the 10 μg group received only a single dose of mRNA-1215 and data only from weeks 0 and 4 are shown. Black vertical arrows represent mRNA-1215 vaccine administration time points. Horizontal dotted line (**e, f**) indicates assay lowest dilution (LOD). ^The exact p-values for all between the group comparisons are presented in Supplementary Tables 5–7.

**Extended Data Table 1 | Geometric Mean NiV(M) Pre-F Binding Antibody Titers (GMTs) and 95% Confidence Intervals (CIs)**

| Timepoint (Week) | Dose of mRNA-1215 | | | |
|---|---|---|---|---|
| | 10 mcg | 25 mcg | 50 mcg | 100 mcg |
| | GMT [95% CIs] IU/mL, N | | | |
| 0 | 3.03 [1.94,4.74] N:10 | 4.44 [1.98,9.95] N:10 | 1.28 [0.83,1.97] N:10 | 1.83 [1.2,2.8] N:10 |
| 1 | 3.11 [1.61,6.01] N:10 | 5.38 [2.04,14.23] N:10 | 2.05 [1.28,3.28] N:10 | 3.56 [1.67,7.55] N:10 |
| 2 | 54.4 [21.81,135.7] N:10 | 119.68 [35.71,401.17] N:10 | 115.72 [68.48,195.54] N:10 | 80.17 [27.72,231.89] N:10 |
| 4 | 109.25 [48.39,246.64] N:10 | 202.88 [71.32,577.13] N:10 | 261.52 [138.16,495.03] N:10 | 146.1 [65.51,325.8] N:10 |
| 5 | 757.15 [410.76,1395.67] N:9 | 1630.04 [861.03,3085.87] N:10 | 2008.24 [999.72,4034.15] N:10 | 1583.71 [1075.9,2331.18] N:10 |
| 6 | 923.47 [526.95,1618.34] N:9 | 1941.62 [1168,3227.66] N:9 | 1885.99 [1025.75,3467.66] N:10 | 1411.64 [1008.13,1976.66] N:10 |
| 8 | 605.76 [316.96,1157.71] N:9 | 1296.59 [797.89,2107.01] N:10 | 1123.63 [422.86,2985.69] N:10 | 1128.36 [785.72,1620.44] N:10 |
| 12 | 400.32 [198.68,806.59] N:9 | 881.24 [507.78,1529.35] N:10 | 1110.21 [614.86,2004.61] N:10 | 853.27 [507.7,1434.06] N:10 |
| 16 | 297.2 [160.5,550.34] N:9 | 647.62 [344.32,1218.08] N:10 | 833.79 [446.7,1556.32] N:10 | 676.44 [392.22,1166.64] N:10 |
| 24 | 164.97 [84.25,323.03] N:9 | 441.2 [226.62,858.96] N:10 | 528.86 [272.6,1026.03] N:10 | 417.44 [223.05,781.26] N:10 |
| 32 | 91.94 [45.04,187.69] N:9 | 304.17 [148.85,621.53] N:10 | 388.73 [206.12,733.09] N:10 | 289.55 [145.46,576.37] N:10 |
| 44 | 75.08 [38.95,144.72] N:9 | 218.71 [112.33,425.85] N:10 | 329.75 [186.73,582.33] N:9 | 195.77 [97.61,392.67] N:10 |
| 56 | 50.53 [22.91,111.44] N:9 | 178.96 [89.82,356.59] N:10 | 253.8 [146.2,440.59] N:9 | 146.43 [71.68,299.11] N:10 |

**Extended Data Table 2 | Geometric Mean NiV(M) G Binding Antibody Titers (GMTs) and 95% Confidence Intervals (CIs)**

| Timepoint (Week) | Dose of mRNA-1215 | | | |
|---|---|---|---|---|
| | 10 mcg | 25 mcg | 50 mcg | 100 mcg |
| | GMT [95% CIs] IU/mL, N | | | |
| 0 | 6.91 [4.14,11.54] N:10 | 12.48 [4.93,31.57] N:10 | 2.84 [1.72,4.68] N:10 | 4.12 [2.48,6.86] N:10 |
| 1 | 7.17 [4.18,12.3] N:10 | 10.86 [3.83,30.81] N:10 | 3.39 [2.23,5.15] N:10 | 5.13 [3.23,8.15] N:10 |
| 2 | 144.98 [69.25,303.52] N:10 | 319.42 [96.31,1059.39] N:10 | 243.72 [115.69,513.42] N:10 | 172.3 [53.79,551.88] N:10 |
| 4 | 304.94 [143.62,647.45] N:10 | 542.73 [222.02,1326.73] N:10 | 663.84 [336.1,1311.18] N:10 | 511.58 [264.96,987.75] N:10 |
| 5 | 1942.66 [1017.61,3708.65] N:9 | 3743.4 [2141.57,6543.35] N:10 | 4944.42 [2493.42,9804.72] N:10 | 4390.7 [3139.16,6141.22] N:10 |
| 6 | 2403.1 [1311.35,4403.76] N:9 | 4626.43 [2962.81,7224.18] N:9 | 5013 [2717.16,9248.69] N:10 | 4279.39 [3293.62,5560.2] N:10 |
| 8 | 1531.68 [777.94,3015.73] N:9 | 3251.82 [2163.31,4888.05] N:10 | 3056.04 [1342.66,6955.86] N:10 | 3385.6 [2679.61,4277.63] N:10 |
| 12 | 913.78 [449.45,1857.82] N:9 | 2149.75 [1373.48,3364.77] N:10 | 2631.35 [1451.35,4770.72] N:10 | 2582.44 [1790.37,3724.92] N:10 |
| 16 | 659.21 [352.05,1234.36] N:9 | 1601.31 [971.69,2638.89] N:10 | 2051.13 [1068.98,3935.65] N:10 | 2108.28 [1403.06,3167.98] N:10 |
| 24 | 369.91 [192.8,709.72] N:9 | 1081.7 [646.56,1809.68] N:10 | 1360.76 [654.22,2830.33] N:10 | 1463.42 [924.66,2316.09] N:10 |
| 32 | 220.58 [122.69,396.54] N:9 | 758.27 [426.89,1346.89] N:10 | 1012.53 [466.84,2196.06] N:10 | 1091.95 [662.7,1799.23] N:10 |
| 44 | 199.24 [103.41,383.89] N:9 | 549.72 [316.11,955.97] N:10 | 952.7 [494.38,1835.9] N:9 | 814.51 [422.89,1568.78] N:10 |
| 56 | 144.68 [65.29,320.6] N:9 | 494.02 [277.47,879.57] N:10 | 769.2 [419.5,1410.42] N:9 | 664.87 [315.02,1403.25] N:10 |

**Extended Data Table 3 | Geometric Mean NiV(M) Neutralizing Antibody Titers (GMTs) and 95% Confidence Intervals (CIs)**

| Timepoint (Week) | Dose of mRNA-1215 | | | |
|---|---|---|---|---|
| | 10 mcg | 25 mcg | 50 mcg | 100 mcg |
| | GMT [95% CIs] IU/mL, N | | | |
| 0 | 8.28 [NA, NA] N:10 | 8.28 [NA, NA] N:10 | 8.28 [NA, NA] N:10 | 8.28 [NA, NA] N:10 |
| 1 | 8.28 [NA, NA] N:10 | 8.28 [NA, NA] N:10 | 8.28 [NA, NA] N:10 | 8.28 [NA, NA] N:10 |
| 2 | 59.58 [33.39,106.34] N:10 | 133.47 [38.95,457.37] N:10 | 113.04 [59.62,214.31] N:10 | 63.91 [23.94,170.61] N:10 |
| 4 | 120.43 [58.58,247.57] N:10 | 204.76 [79.57,526.91] N:10 | 261.84 [122.79,558.36] N:10 | 205.43 [89.55,471.3] N:10 |
| 5 | 1239.53 [607.81,2527.82] N:9 | 2598.31 [1511.79,4465.71] N:10 | 3641.44 [1659.44,7990.73] N:10 | 3520.92 [2094.01,5920.14] N:10 |
| 6 | 1491.37 [828.52,2684.55] N:9 | 3118.27 [1887.14,5152.54] N:9 | 3686.81 [1774.92,7658.11] N:10 | 3552.51 [2493.3,5061.71] N:10 |
| 8 | 908.58 [434.93,1898.02] N:9 | 2223.88 [1464.23,3377.64] N:10 | 2008.32 [722.44,5582.98] N:10 | 2590.35 [1747.97,3838.69] N:10 |
| 12 | 489.83 [224.74,1067.62] N:9 | 1413.91[923.69,2164.3] N:9 | 1732.36 [827.06,3628.61] N:10 | 1838.87 [1222.52,2765.97] N:10 |
| 16 | 324.33 [145.71,721.95] N:9 | 1037.37 [610.68,1762.18] N:10 | 1181.24 [525.83,2653.56] N:10 | 1211.44 [759.78,1931.61] N:10 |
| 24 | 148.78 [61.2,361.7] N:9 | 491.27 [284.46,848.45] N:10 | 627.39 [264.22,1489.76] N:10 | 697.63 [378.2,1286.86] N:10 |
| 32 | 85.75 [42.57,172.75] N:9 | 289.09 [162.42,514.54] N:10 | 396.3 [172.42,910.88] N:10 | 397.02 [212.58,741.51] N:10 |
| 44 | 70.39 [34.3,144.48] N:9 | 178.59 [98.29,324.49] N:10 | 310.89 [133.83,722.2] N:9 | 259.49 [137.75,488.83] N:10 |
| 56 | 63.4 [30.19,133.14] N:9 | 150.65 [81.14,279.69] N:10 | 244.76 [111.34,538.06] N:9 | 171.07 [82.6,354.32] N:10 |

**Extended Data Table 4 | Geometric Mean NiV(B) Neutralizing Antibody Titers (GMTs) and 95% Confidence Intervals (CIs)**

| Timepoint (Week) | Dose of mRNA-1215 | | | |
|---|---|---|---|---|
| | 10 mcg | 25 mcg | 50 mcg | 100 mcg |
| | GMT [95% CIs] IU/mL, N | | | |
| 0 | 4.73 [NA, NA] N:10 | 4.73 [NA, NA] N:10 | 4.73 [NA, NA] N:10 | 4.73 [NA, NA] N:10 |
| 4 | 31.03 [14.86,64.78] N:10 | 65.16 [24.89,170.62] N:10 | 68.91 [32.88,144.42] N:10 | 47.69 [20.54,110.7] N:10 |
| 6 | 599.17 [294.16,1220.47] N:9 | 1318.7 [678.29,2563.77] N:9 | 1252.96 [687.05,2285] N:10 | 1023.54 [617.44,1696.73] N:10 |
| 12 | 218.18 [109.36,435.28] N:9 | 508.99 [288.19,898.95] N:10 | 628.51 [297.58,1327.48] N:10 | 667.59 [392.71,1134.87] N:10 |
| 24 | 72.36 [37.79,138.55] N:9 | 283.76 [134.03,600.79] N:10 | 318.13 [151.87,666.39] N:10 | 290.18 [133.55,630.52] N:10 |
| 56 | 36.87 [16.39,82.96] N:10 | 87.68 [40.8,188.45] N:10 | 148.82 [67.77,326.83] N:9 | 89.29 [38.7,205.99] N:10 |

**Extended Data Table 5 | Geometric Mean HeV Pre-F and G Binding Antibody Titers (GMTs) and 95% Confidence Intervals (CIs)**

| Timepoint (Week) | Dose of mRNA-1215 | | | |
|---|---|---|---|---|
| | 10 mcg | 25 mcg | 50 mcg | 100 mcg |
| | GMT [95% CIs] EC50, N | | | |
| | **Pre-F Binding Antibody Titers** | | | |
| 0 | 2.79 [1.71,4.54] N:10 | 4.53 [2.44,8.39] N:10 | 1.82 [1.08,3.06] N:10 | 2.64 [1.47,4.73] N:10 |
| 4 | 60.52 [28.71,127.57] N:10 | 104.94 [47.15,233.54] N:10 | 137.77 [77.17,245.95] N:10 | 111.36 [57.61,215.28] N:10 |
| 6 | 598.6 [342.71,1045.56] N:9 | 1074.5 [665.99,1733.58] N:9 | 1214.22 [660.75,2231.27] N:10 | 1260.33 [949.38,1673.13] N:10 |
| 12 | 241.02 [130.29,445.83] N:9 | 583.74 [405.46,840.41] N:10 | 708.43 [390.56,1285.03] N:10 | 818.47 [558.47,1199.52] N:10 |
| 24 | 103.93 [54.58,197.9] N:9 | 332.19 [202.47,545.03] N:10 | 390.59 [186.94,816.1] N:10 | 430.22 [252.79,732.2] N:10 |
| 56 | 49.4 [16.64,146.69] N:9 | 167.33 [91.39,306.36] N:10 | 197.65 [92.74,421.23] N:9 | 197.03 [82.28,471.82] N:10 |
| | **G Binding Antibody Titers** | | | |
| 0 | 2.79 [1.71,4.54] N:10 | 4.53 [2.44,8.39] N:10 | 1.82 [1.08,3.06] N:10 | 2.64 [1.47,4.73] N:10 |
| 4 | 60.52 [28.71,127.57] N:10 | 104.94 [47.15,233.54] N:10 | 137.77 [77.17,245.95] N:10 | 111.36 [57.61,215.28] N:10 |
| 6 | 598.6 [342.71,1045.56] N:9 | 1074.5 [665.99,1733.58] N:9 | 1214.22 [660.75,2231.27] N:10 | 1260.33 [949.38,1673.13] N:10 |
| 12 | 241.02 [130.29,445.83] N:9 | 583.74 [405.46,840.41] N:10 | 708.43 [390.56,1285.03] N:10 | 818.47 [558.47,1199.52] N:10 |
| 24 | 103.93 [54.58,197.9] N:9 | 332.19 [202.47,545.03] N:10 | 390.59 [186.94,816.1] N:10 | 430.22 [252.79,732.2] N:10 |
| 56 | 49.4 [16.64,146.69] N:9 | 167.33 [91.39,306.36] N:10 | 197.65 [92.74,421.23] N:9 | 197.03 [82.28,471.82] N:10 |

**Extended Data Table 6 | Geometric Mean HeV Neutralizing Antibody Titers (GMTs) and 95% Confidence Intervals (CIs)**

| Timepoint (Week) | Dose of mRNA-1215 | | | |
|---|---|---|---|---|
| | 10 mcg | 25 mcg | 50 mcg | 100 mcg |
| | GMT [95% CIs] ID50, N | | | |
| 0 | 25 [NA, NA] N:10 | 25 [NA, NA] N:10 | 25 [NA, NA] N:10 | 25 [NA, NA] N:10 |
| 4 | 50.97 [31,83.8] N:10 | 86.41 [34.67,215.35] N:10 | 101.07 [49.44,206.61] N:10 | 70 [32.12,152.55] N:10 |
| 6 | 868.51 [438.25,1721.18] N:9 | 2132.15 [1277.86,3557.55] N:9 | 2013.09 [1046.15,3873.78] N:10 | 1791.92 [1261.91,2544.53] N:10 |
| 12 | 290.72 [148.71, 568.37] N:9 | 954.02 [561.56,1620.76] N:10 | 1047.9 [502.04,2187.28] N:10 | 1213.03 [687.56,2140.1] N:10 |
| 24 | 130.75 [61.37,278.57] N:9 | 497.81 [251.11,986.87] N:10 | 564.42 [235.67,1351.76] N:10 | 533.19 [254.24,1118.22] N:10 |
| 56 | 83.26 [39.35,176.18] N:9 | 187.77 [81.74,431.32] N:10 | 239.87 [90.94,632.71] N:9 | 170 [79.89,361.74] N:10 |

# Reporting Summary

## Statistics

For all statistical analyses, confirm that the following items are present in the figure legend, table legend, main text, or Methods section.

| n/a | Confirmed | |
|---|---|---|
| ☐ | ☒ | The exact sample size (*n*) for each experimental group/condition, given as a discrete number and unit of measurement |
| ☐ | ☒ | A statement on whether measurements were taken from distinct samples or whether the same sample was measured repeatedly |
| ☐ | ☒ | The statistical test(s) used AND whether they are one- or two-sided<br>*Only common tests should be described solely by name; describe more complex techniques in the Methods section.* |
| ☐ | ☒ | A description of all covariates tested |
| ☐ | ☒ | A description of any assumptions or corrections, such as tests of normality and adjustment for multiple comparisons |
| ☐ | ☒ | A full description of the statistical parameters including central tendency (e.g. means) or other basic estimates (e.g. regression coefficient) AND variation (e.g. standard deviation) or associated estimates of uncertainty (e.g. confidence intervals) |
| ☐ | ☒ | For null hypothesis testing, the test statistic (e.g. *F*, *t*, *r*) with confidence intervals, effect sizes, degrees of freedom and *P* value noted<br>*Give P values as exact values whenever suitable.* |
| ☒ | ☐ | For Bayesian analysis, information on the choice of priors and Markov chain Monte Carlo settings |
| ☒ | ☐ | For hierarchical and complex designs, identification of the appropriate level for tests and full reporting of outcomes |
| ☐ | ☒ | Estimates of effect sizes (e.g. Cohen's *d*, Pearson's *r*), indicating how they were calculated |

*Our web collection on statistics for biologists contains articles on many of the points above.*

## Software and code

Policy information about availability of computer code

| Data collection | ELISA data were collected using an Envision plate reader (PerkinElmer); neutralization activity was measured using SpectraMax L luminometer (Molecular Devices); B and T cell data were acquired using BD FACSymphony cytometer; Clinical data : AdvantageEDC (SM), Regulatory Tracking System (RTS) maintained and hosted by the Clinical Program Support Center (CPSC) at the EMMES Corporation. |
|---|---|
| Data analysis | R version 4.3.3., mmrm package in R; Prism, versions 10.0.0, 10.2.2, and 10.4.1, FlowJo version 10.10.0. |

For manuscripts utilizing custom algorithms or software that are central to the research but not yet described in published literature, software must be made available to editors and reviewers. We strongly encourage code deposition in a community repository (e.g. GitHub). See the Nature Portfolio guidelines for submitting code & software for further information.

## Data

Policy information about availability of data

All manuscripts must include a data availability statement. This statement should provide the following information, where applicable:

- Accession codes, unique identifiers, or web links for publicly available datasets
- A description of any restrictions on data availability
- For clinical datasets or third party data, please ensure that the statement adheres to our policy

Results generated in this study are available as de-identified data on ClinicalTrials.gov. https://clinicaltrials.gov/study/NCT05398796?cond=Nipah&term=vaccine&rank=2&a=51&b=52. The study protocol and informed consent form are available on ClinicalTrials.gov (https://clinicaltrials.gov/study/

# Research involving human participants, their data, or biological material

Policy information about studies with [human participants or human data](). See also policy information about [sex, gender (identity/presentation), and sexual orientation]() and [race, ethnicity and racism]().

| | |
|---|---|
| Reporting on sex and gender | Participants self-reported sex on the day of the enrollment and these results are reported in Supplementary Table 1. No sex or gender based analyses were performed as this was not an objective of the trial. Such investigation is beyond the scope of these small phase 1 clinical trials and should be further investigated in the subsequent clinical investigation phases. |
| Reporting on race, ethnicity, or other socially relevant groupings | Participants self-reported on their race and ethnicity. Participants chose their race from options including Asian, Black or African American, White, and Multiracial and their ethnicity as either Non-Hispanic or Latino or Hispanic/Latino. For both race and ethnicity, it was permitted to decline self-reporting of these categories, in which case participants are reported as Unknown/Not reported. |
| Population characteristics | Eligible participants were adults 18 to 60 years of age who were in good general health, as determined by medical history, physical examination, and laboratory testing. Exclusion criteria related to the pathogen target of the mRNA-1215 vaccine were confirmed past Nipah virus infection or prior residence for greater than 6 months or planned travel during the study for any length of time to places where Nipah virus infection is endemic. Forty participants, 18 females (45%) and 22 males (55%), with an overall mean age of 37 (range 22 to 59) were enrolled into the study from July 11, 2022 to August 22, 2023 |
| Recruitment | Participants were recruited from the greater Washington, D.C. area using IRB-approved recruitment ads. the trial was open-labeled and non-randomized, which could potentially introduce an element of bias in the reporting of reactogenicity. |
| Ethics oversight | This phase I clinical trial was reviewed and approved by the NIH Institutional Review Board (IRB). |

Note that full information on the approval of the study protocol must also be provided in the manuscript.

# Field-specific reporting

Please select the one below that is the best fit for your research. If you are not sure, read the appropriate sections before making your selection.

☒ Life sciences        ☐ Behavioural & social sciences        ☐ Ecological, evolutionary & environmental sciences

For a reference copy of the document with all sections, see [nature.com/documents/nr-reporting-summary-flat.pdf]()

# Life sciences study design

All studies must disclose on these points even when the disclosure is negative.

| | |
|---|---|
| Sample size | Sample size determination for this trial was based on the primary endpoint of safety. These calculations for safety were predetermined and expressed in terms of the ability to detect SAEs. Sample sizes were chosen so that there was a 90% chance to observe at least one SAE if the true rate was at least 0.21, and over a 90% chance to observe no SAE if the true rate was no more than 0.01. The study was not designed to detect large immunologic differences between the groups (i.e. 1.2 times the standard deviation of the immune response with 80% power). Adjustments for multiple comparisons were not performed. |
| Data exclusions | One participant in the 10 mcg group received only a single dose of mRNA-1215 and data collected only at weeks 0, 2, and 4 was included in the analysis for this participant. No other data was excluded. |
| Replication | Each ELISA assay was done in triplicate with duplicate values inside each assay. A plasma positive control present on each plate was used to determine if the assay passed or failed according to acceptance criteria pre-established in advance; only passed assays were analyzed and reported. Five plasma bridge controls, included for each assay, were also used to assess reproducibility across assays. For measured samples, from the triplicate assays, any sample with a CV>30% across the 3 EC50 values determined was rerun in a 4th assay and that 4th value was included in the final analysis.Neutralization assays were run in duplicate or triplicate with control antibody samples included in each plate for reproducibility. |
| Randomization | This trial was non-randomized. The study group assignment was set up in the database prior to opening the study to accrual. The group assignment was known to the staff and study participants before completing the electronic enrollment into the study on Day 0. The assignment to a group was not randomized because this is not crucial for descriptive and exploratory research that is not evaluating a treatment effect as would occur in a phase 2 or 3 study. |
| Blinding | We did not include blinding in this trial, because it was not needed to assess safety and tolerability or immunogenicity in this first-in-human trial of a novel vaccine administered to a naïve population. |

# Reporting for specific materials, systems and methods

We require information from authors about some types of materials, experimental systems and methods used in many studies. Here, indicate whether each material, system or method listed is relevant to your study. If you are not sure if a list item applies to your research, read the appropriate section before selecting a response.

### Materials & experimental systems

| n/a | Involved in the study |
|---|---|
| ☐ | ☒ Antibodies |
| ☐ | ☒ Eukaryotic cell lines |
| ☒ | ☐ Palaeontology and archaeology |
| ☒ | ☐ Animals and other organisms |
| ☐ | ☒ Clinical data |
| ☒ | ☐ Dual use research of concern |
| ☒ | ☐ Plants |

### Methods

| n/a | Involved in the study |
|---|---|
| ☒ | ☐ ChIP-seq |
| ☐ | ☒ Flow cytometry |
| ☒ | ☐ MRI-based neuroimaging |

## Antibodies

| Antibodies used | ELISA: anti-non-human primate IgG-horseradish peroxidase conjugate cross-reacting to human IgG, Southern Biotech Catalogue number #2040-05-Lot E0922-Z742C expiration date 2024-06; Lot C3923-XJ83C expiration date 2025-07<br><br>T cell analysis:<br>1. Live/dead fixable aqua dead cell stain, Invitrogen #L34957 – Lot #2696857[1:800]<br>2. CD45RA PE-CY5, clone 5H9, BD Biosciences #552888 – Lot #1118897[1:2500]<br>3. CD4 PE-CY5.5, clone SK3, Thermo Fisher #35-0047-42 – Lot #2521213[1:2500]<br>4. ICOS PE-CY7, clone C398.4A, Biolegend #313520 – Lot #B293719[1:640]<br>5. CD8 BV570, clone RPA-T8, Biolegend #301038 – Lot #B367805[1:80]<br>6. CCR7 BV650, clone GO43H7, Biolegend #353234 – Lot #B370728[1:10]<br>7. CXCR3 BV711, clone 1C6/CXCR3, BD Biosciences #563156 – Lot #2129036[1:20]<br>8. PD-1 BUV737, clone EH12.1, BD Horizon #612792 – Lot #3220435[1:20]<br>9. TNF FITC, clone Mab11, BD Biosciences #554512 – Lot #2213980[1:80]<br>10. IL-4 BB700, clone MP4-25D2, BD Biosciences custom order – Lot #1145122[1:20]<br>11. CXCR5 PE, clone MU5UBEE, Thermo Fisher #12-9185-42 – Lot #2404260[1:10]<br>12. CD69 ECD, clone TP1.55.3, Beckman Coulter #7620104 [1:40]<br>13. IL-21 Ax647, clone 3A3-N2.1, BD Biosciences #560493 – Lot #2266971[1:10]<br>14. IFN-g Ax700, clone B27, Biolegend #506516 – Lot #B320892[1:640]<br>15. CD3 APC-CY7, clone SP34.2, BD Biosciences #557757 – Lot #1152687[1:320]<br>16. IL-13 BV421, clone JES10-5A2, BD Biosciences #563580 – Lot #3086969[1:20]<br>17. IL-17A BV605, clone BL168, Biolegend #512326 – Lot #B376461[1:20]<br>18. CD154 BV785, clone 24-31, Biolegend #310842 – Lot #B329207[1:20]<br>19. IL-2 BV750, clone MQ1-17H12, BD Biosciences #566361 – Lot #2285235[1:40]<br><br>B cell analysis:<br>1. Live/dead fixable aqua dead cell stain, Invitrogen #L34957 – Lot #2696857[1:800]<br>2. IgD FITC, goat pAb, Southern Biotech #2030-02 – Lot #A2118-WF09C [1:160]<br>3. IgM PerCP-Cy5.5, clone G20-127, BD Biosciences #561285 – Lot #0307134[1:40]<br>4. IgA Dy405, goat pAb, Jackson ImmunoResearch #109-475-011 – Lot #150866 [1:80]<br>5. CD20 BV570, clone 2H7, Biolegend #302332 – Lot #B301458[1:40]<br>6. CD27 BV650, clone O323, Biolegend #302828 – Lot #B350350[1:20]<br>7. CD14 BV785, clone M5E2, Biolegend #301840 – Lot #B327948 [1:80]<br>8. CD16 BUV496, clone 3G8, BD Biosciences #564653 – Lot #0155949[1:80]<br>9. CD4 BUV737, clone SK3, BD Biosciences #564305 – Lot #0282762[1:320]<br>10. CD19 APC, clone J3-119, Beckman Coulter #IM2470U – Lot #200093[1:80]<br>11. IgG Ax700, clone G18-145, BD Biosciences #561296 – Lot #0135021[1:20]<br>12. CD3 APC-Cy7, clone SP34.2, BD Biosciences #557757 – Lot #1152687[1:40]<br>13. CD38 PE, clone OKT10, Caprico Biotech #100826 – Lot #8AE4[1:160]<br>14. CD21 PE-Cy5, clone B-ly4, BD Biosciences #551064 – Lot #0072939[1:40]<br>15. CXCR5 PE-Cy7, clone MU5UBEE, Thermo Fisher #25-9185-42 – Lot #2442267[1:40] |
|---|---|
| Validation | All antibodies were commercially manufactured and validated. ELISA: Each lot of antibody was thoroughly bridged to match historical potency and reproducibility with a series of plasma samples from both non-human primate and human origin. In such the dilution at which each antibody was used differed from lot-to-lot. All antibodies for flow cytometry were titrated on human PBMC and the concentrations that gave the best separation were selected. Representative staining is shown in the gating trees for the Intracellular Cytokine Staining and B cell panels (Extended Data Figure 2,3). |

# Eukaryotic cell lines

Policy information about cell lines and Sex and Gender in Research

| | |
|---|---|
| Cell line source(s) | HEK293T (ATCC #CRL-11268) is a cell line exhibiting epithelial morphology that was isolated from human embryo kidney tissue. |
| Authentication | Since the HEK293Tcell line was sourced from ATCC, independent authentication was not done. ATCC characterization report can be found here https://www.atcc.org/products/crl-11268 |
| Mycoplasma contamination | Cell lines were not tested for mycoplasma contamination. |
| Commonly misidentified lines (See ICLAC register) | None |

# Clinical data

Policy information about clinical studies

All manuscripts should comply with the ICMJE guidelines for publication of clinical research and a completed CONSORT checklist must be included with all submissions.

| | |
|---|---|
| Clinical trial registration | NCT05398796 |
| Study protocol | The protocol is available in ClinicalTrials.gov under the corresponding trial registration number, as well as in the Supplementary Appendix for this manuscript. |
| Data collection | Study participants were enrolled into the study from July 11, 2022 to August 22, 2023 at the Vaccine Evaluation Clinic in the National Institutes of Health Clinical Center. Data were collected through September 17, 2024. |
| Outcomes | The primary objective of the trial was to evaluate the safety and tolerability of a 2-dose vaccination regimen of mRNA-1215 at doses of 10 mcg, 25 mcg, 50 mcg or 100 mcg administered IM, given at a 4-week interval. For safety monitoring, all study participants were observed for a minimum of 30 minutes following each vaccination. Vital signs (temperature, blood pressure, pulse and respiratory rate) and assessment of local reactogenicity were performed after each product administration. Participants reported solicited local and systemic reactogenicity for the first 7 days following each vaccination. Adverse events (AEs) were collected for the first 28 days after each vaccination, while serious adverse events (SAEs) and new chronic medical conditions were recorded throughout the trial. The secondary objective of the trial was to evaluate antibody responses to the mRNA-1215 vaccine at doses of 10 mcg, 25 mcg, 50 mcg or 100 mcg at 2 weeks after last product administration. Serum samples were collected at protocol-specified timepoints for immunogenicity analysis of vaccine-induced antibody responses by Nipah virus IgG enzyme-linked immunosorbent assay (ELISA). |

# Plants

| | |
|---|---|
| Seed stocks | *Report on the source of all seed stocks or other plant material used. If applicable, state the seed stock centre and catalogue number. If plant specimens were collected from the field, describe the collection location, date and sampling procedures.* |
| Novel plant genotypes | *Describe the methods by which all novel plant genotypes were produced. This includes those generated by transgenic approaches, gene editing, chemical/radiation-based mutagenesis and hybridization. For transgenic lines, describe the transformation method, the number of independent lines analyzed and the generation upon which experiments were performed. For gene-edited lines, describe the editor used, the endogenous sequence targeted for editing, the targeting guide RNA sequence (if applicable) and how the editor was applied.* |
| Authentication | *Describe any authentication procedures for each seed stock used or novel genotype generated. Describe any experiments used to assess the effect of a mutation and, where applicable, how potential secondary effects (e.g. second site T-DNA insertions, mosiacism, off-target gene editing) were examined.* |

# Flow Cytometry

## Plots

Confirm that:

☒ The axis labels state the marker and fluorochrome used (e.g. CD4-FITC).

☒ The axis scales are clearly visible. Include numbers along axes only for bottom left plot of group (a 'group' is an analysis of identical markers).

☒ All plots are contour plots with outliers or pseudocolor plots.

☒ A numerical value for number of cells or percentage (with statistics) is provided.

## Methodology

| | |
|---|---|
| Sample preparation | B cells: Cryopreserved PBMC were thawed, washed briefly with phenol-free RPMI/4% heat inactivated newborn calf serum (R&D Systems) and incubated with aqua live/dead fixable dead cell stain kit (Thermo Fisher Scientific) for 20 minutes at RT. |

Cells were stained with the following antibodies (monoclonal unless indicated) for 20 minutes at RT: IgD FITC [1:160] (goat polyclonal, Southern Biotech), IgM PerCP-Cy5.5 [1:20] (clone G20-127, BD Biosciences), IgA Dylight 405 [1:80] (goat polyclonal, Jackson Immunoresearch Inc), CD20 BV570 [1:40] (clone 2H7, Biolegend), CD27 BV650 [1:20] (clone O323, Biolegend), CD14 BV785 [1:80] (clone M5E2, Biolegend), CD16 BUV496 [1:80] (clone 3G8, BD Biosciences), CD4 BUV737 [1:320] (clone SK3, BD Biosciences), CD19 APC [1:80] (clone J3-119, Beckman), IgG Alexa 700 [1:20] (clone G18-145, BD Biosciences), CD3 APC-Cy7 [1:40] (clone SP34-2, BD Biosciences), CD38 PE [1:160] (clone OKT10, Caprico Biotechnologies), CD21 PE-Cy5 [1:40] (clone B-ly4, BD Biosciences) and CXCR5 PE-Cy7 [1:40] (clone MU5UBEE, Thermo Fisher Scientific). Stained cells were then incubated with matched Pre-F trimer and G trimer probe pairs: streptavidin-BUV661 (BD Biosciences) labeled Bangladesh Pre-F probe [1:25] and streptavidin-BUV395 labeled Malaysia Pre-F probe [1:25], and streptavidin-BUV661 (BD Biosciences) labeled Bangladesh G probe [1:25] and streptavidin-BUV395 labeled Malaysia G probe [1:25] for 30 minutes at 4°C (protected from light). The Pre-F probe used in these assays (GenBank AJ627196) is derived from a NiV Malaysia 1999 isolate that differs by only one amino acid, at position T348M, from AJ564621, the isolate sequence used for mRNA-1215. The NiV Malaysian G probe matched the sequence of G in the vaccine. For the Bangladesh probes, both the Pre-F and G amino acid sequences were derived from the 2004 sequence Genebank MK673564. Cells were washed and fixed in 0.5% formaldehyde (Tousimis Research Corp) prior to data acquisition.

T cells: Cryopreserved PBMCs were thawed and incubated overnight in at 37°C/5% CO2. After incubation, cells were stimulated with Malaysia Pre-F or Malaysia G peptide pools [Malaysia 1999 isolate NV/MY/99/VRI-2794 (GenBank AJ564621); JPT Peptides] at a final concentration of 2 μg/ml in the presence of 3 mM monensin for 6 hours. The Pre-F and G peptide pools are comprised of 123 and 108 individual peptides, respectively, as 15mers overlapping by 11 amino acids in 100% DMSO.  Negative controls received an equal concentration of DMSO to that of peptide pools (final concentration of 0.5%). Cells were stained with the following monoclonal antibodies: CD3 APC-Cy7 [1:160] (clone SP34.2, BD Biosciences), CD4 PE-Cy5.5 [1:80] (clone SK3, Thermo Fisher), CD8 BV570 [1:40] (clone RPA-T8, BioLegend), CD45RA PE-Cy5 [1:2500] (clone 5H9, BD Biosciences), CCR7 BV650 [1:10] (clone G043H7, BioLegend), CXCR5 PE [1:10] (clone MU5UBEE, Thermo Fisher), CXCR3 BV711 [1:20] (clone 1C6/CXCR3, BD Biosciences), PD-1 BUV737 [1:20] (clone EH12.1, BD Biosciences), ICOS Pe-Cy7 [1:80] (clone C398.4A, BioLegend), CD69 ECD [1:40] (cloneTP1.55.3, Beckman Coulter), IFN-g Ax700 [1:320] (clone B27, BioLegend), IL-2 BV750 [1:40] (clone MQ1-17H12, BD Biosciences), IL-4 BB700 [1:20] (clone MP4-25D2, BD Biosciences), TNF-FITC [1:80] (clone Mab11, BD Biosciences), IL-13 BV421 [1:20] (clone JES10-5A2, BD Biosciences), IL-17 BV605 [1:20] (clone BL168, BioLegend), IL-21 Ax647 [1:10] (clone 3A3-N2.1, BD Biosciences), and CD154 BV785 [1:20] (clone 24-31, BioLegend). Aqua live/dead fixable dead cell stain kit (Thermo Fisher Scientific) was used to exclude dead cells.

| | |
|---|---|
| Instrument | Samples were acquired on an BD FACSymphony flow cytometer. |
| Software | Samples were analyzed  using FlowJo version 10.10.0 (Treestar, Inc., Ashland, OR). |
| Cell population abundance | N/A |
| Gating strategy | B cell gating strategy is shown in Extended Data Figure 2; T cell gating strategy is shown in Extended Data Figure 3. |

☒ Tick this box to confirm that a figure exemplifying the gating strategy is provided in the Supplementary Information.

