## [Peer Review File · Nature Medicine]

A structure-based mRNA vaccine for Nipah Virus in healthy adults: a phase 1 trial

Corresponding Author: Dr Lesia Dropulic

Version 0:

Reviewer comments:

Reviewer #1

(Remarks to the Author)

This phase-1, open-label, dose-escalation study of the mRNA-1215 Nipah-virus vaccine offers valuable first-in-human safety and immunogenicity data. The design is appropriate for an early-stage trial, yet several statistical issues, principally around precision, multiplicity, missing data and longitudinal modelling, should be addressed.

1 Study design and population

Cohort size (n = 10 per dose). Therefore, 1: Limits the precision of adverse-event (AE) and immune-response estimates. 2: Makes formal hypothesis testing essentially uninformative; results should be framed as exploratory.

2 Statistical methodology

2.1 Descriptive statistics with exact 95 % CIs for proportions and geometric mean titres (GMTs) are sensible, but the manuscript must state the statistical software and packages used.

2.2 P-values are reported for multiple secondary endpoints across doses and time points, yet no adjustment is planned. control the family-wise error rate (e.g., Holm) or false-discovery rate.

2.3 Complete-case analyses assume data are missing completely at random—unlikely over 56 weeks.

2.4 Both intent-to-treat (ITT) and per-protocol sets are defined but only ITT results are shown. Present both or justify their merger, noting at least one participant missed dose 2.

2.5 Time-course GMTs are plotted but not modelled. A mixed-effects model (log-titre ~ dose × time, random intercepts) would should be considered to exploit within-subject correlation.

3 Results presentation

3.1 Add 95 % CIs to every figure/table (some Extended Data panels show raw counts only).

3.2 Improve figure accessibility—use distinct shapes or patterns, not just colours, to accommodate colour-blind readers.

4 Interpretation and generalisability

4.1 Immunogenicity is compared with convalescent sera collected > 20 years post-infection; differences in age, waning immunity and assay technology weaken that benchmark.

4.2 Statements of “dose independence” should be tempered; with ten participants per group, only extreme differences are detectable.

4.3 Outline how these phase-1 data map to surrogate endpoints acceptable under accelerated or Animal-Rule regulatory pathways.

5 Minor clarifications

5.1 Specify whether GMT CIs are exact or asymptotic (back-transformed log-means?).

5.2 Describe how values below the lower limit of quantification were handled.

Reviewer #3

(Remarks to the Author)

Brief summary and key results:

This manuscript discusses an open-label, phase 1, first-in-human, dose-escalation study designed to evaluate the safety, tolerability, and immunogenicity of a two-dose vaccination regimen of the chimeric Pre-F/G mRNA vaccine, mRNA-1215, in healthy adult volunteers. The vaccine is notable for incorporating the stabilized pre-F antigen, representing a first-in-class approach for clinical testing.

Overall, the study showed that the vaccine was generally safe, with only mild to moderate side effects consistent with prior

experience from mRNA vaccine trials. Though, of note, a single case of prolonged urticaria occurred in the high-dose group, a side effect also observed in some recipients of COVID-19 mRNA vaccines. This reaction required anti-histamine treatment for approximately five months before full resolution.

Administration of two doses, four weeks apart, induced strong antigen-specific antibody titers with demonstrated neutralizing activity. The study commendably followed titers for over a year, providing evidence for the durability of the immune response. Investigators also assessed the breadth of protection by comparing neutralization against both NiV(BM) and NiV(B) strains, as well as cross-reactive responses to HeV, evaluating both binding and neutralizing antibody titers.

General assessment:

The trial was conducted according to protocol, with only one deviation in which one participant received a single dose, as appropriately noted in the report. The data are presented clearly, and the results are thoroughly discussed. Appropriate standards were used as a comparison when possible. Overall, I agree with the conclusion that the findings are favorable to continue the clinical development of mRNA-1215 against Nipah.

Comments:

1. Clarification on B-cell response data. I found the data somewhat confusing. In the panels Figure 4 panels a and b, the frequencies of antigen-specific memory B-cell responses were assessed at various study time points using fluorescently labeled Pre-F and G trimer probes, either matched for both NiV(M) and NiV(B) (top) or specific to NiV(M) antigens (bottom). The top panel shows a clear response for both Pre-F and G, whereas the bottom panel demonstrates no detectable response for Pre-F and only a weak signal for G. Does that mean that the NiV(B) probes account for the apparent difference? This result seems unexpected, given that the vaccine was designed based on NiV(M). Some additional clarification or interpretation from the investigators would be helpful.

2. Lack of details on vaccine formulation. The manuscript provides little information on the vaccine formulation, particularly regarding the lipid composition. I assume that the ratio of mRNA to lipid remains constant across the dose-escalation study, but does this also imply that the lipid dose increases proportionally? Since lipid nanoparticles can influence both immunogenicity and reactogenicity, I would prefer to see the authors add more detail on the formulation as well as a brief discussion on the potential toxicity of the lipid components.

Reviewer #4

(Remarks to the Author)

In this manuscript authors analyzed vaccine safety, tolerability and immunogenicity of the lipid nanoparticle mRNA vaccine against Nipah virus (NiV) in the phase 1, first-in-human, open-label, dose-escalation trial. They immunized intramuscularly 40 volunteers, with mRNA-1215, using 2 doses at a 4-week interval. They observed in immunized individuals reactogenicity symptoms including mild pain and tenderness at the injection site (n=33) and mild malaise (n=16). mRNA-1215 elicited robust NiV F and G binding antibody titers and neutralizing titers by 2 weeks after the prime, remaining detectable elevated for at least one year after vaccination. They suggest mRNA-1215 as a promising vaccine candidate to advance for pandemic preparedness and for those at risk from regional outbreaks. Nipah virus is highly pathogenic virus for which neither licensed vaccine nor therapeutics are currently available, underlying the importance of the development of novel prophylactic approaches, like the one presented in this manuscript.

The study of vaccine safety and tolerability was well performed and results presented correctly. However, all studies of vaccine immunogenicity were performed without any test implicating the live virus. Although the utilization of pseudovirus seroneutralisation assay most often correlates with the neutralization of the live virus, it cannot completely replace it in the clinical trial of the new Nipah vaccine and is indispensable to claim the new vaccine candidate for Nipah. This is even more important as the proof-of-concept study of the mRNA-1215 vaccine in nonhuman primates is still not published, making it difficult to verify results presented only in the meeting abstract (ref. 35 of the manuscript).

Authors claim that "starting 2 weeks after vaccination, the neutralizing antibody titers induced by mRNA-1215 were similar to titers of NiV convalescent patients" (line 218, Fig. 3f). As the sera from convalescent patients were obtained more than 20 years after NiV infection and the antibody titer wanes with the time, this suggest that mRNA-1215-induced neutralizing titer is not very high, which does not seem to be particularly rewarding as a message of the paper?

In the discussion (line 302), authors claim that they are no other published vaccines which target both NiV F and G. The statement should be corrected as there are others published work presenting this type of vaccines (for ex. Pastor Y. et al, 2024).

How do authors explain that CD8 T cell responses were generated following vaccination only against NiV F and not NiV G?

Version 1:

Reviewer comments:

Reviewer #1

(Remarks to the Author)

Thank you for revision. I have no further comment.

Reviewer #3

(Remarks to the Author)

I am satisfied with the changes made by the authors in response to my comments. In addition I found the additional changes made based on the suggestions of the other reviewers to further clarify the data of the clinical study.

Reviewer #4

(Remarks to the Author)

Authors have successfully answered to this reviewer's comments (although the lines marked to contain the modifications to responses to questions 1 and 4 were different in the discussion of modified manuscript).

Reviewer #1 (Remarks to the Author):

This phase-1, open-label, dose-escalation study of the mRNA-1215 Nipah-virus vaccine offers valuable first-in-human safety and immunogenicity data. The design is appropriate for an early-stage trial, yet several statistical issues, principally around precision, multiplicity, missing data and longitudinal modelling, should be addressed.

1. Study design and population

Cohort size (n = 10 per dose). Therefore, 1: Limits the precision of adverse-event (AE) and immune-response estimates. 2: Makes formal hypothesis testing essentially uninformative; results should be framed as exploratory.

Response: We agree that the cohort size (n=10, total n=40) limits the precision of adverse event and immunogenicity estimates and acknowledge that the small number of participants in this phase I study is one of its limitations. We added a statement in the Statistical analysis section (lines 1110-1114) and in the Discussion (lines 474-476) to further clarify this and frame the results in an exploratory context. The primary objective of this trial was to begin to evaluate the safety and tolerability of the two-dose vaccination regimen at each dose, and the secondary objective was to evaluate Pre-F and G antibody responses to the mRNA-1215 vaccine at 2 weeks after the second vaccination at each dose level in a small, first-in-human study. In this context, our study met its intended objectives and will help guide future clinical development of the NiV mRNA-1215 vaccine.

2. Statistical methodology

2.1 Descriptive statistics with exact 95 % CIs for proportions and geometric mean titers (GMTs) are sensible, but the manuscript must state the statistical software and packages used.

Response: All original statistical analyses were performed in R version 4.3.3, as detailed in the Statistical analysis section (line 1145). Per the reviewer's suggestion (as noted in 2.5 below), we added a longitudinal analysis using mixed models for repeated measures. This was done using the mmrm (Mixed Model for Repeated Measures) package in R which is now cited in the manuscript (line 1117).

2.2 P-values are reported for multiple secondary endpoints across doses and time points, yet no adjustment is planned. control the family-wise error rate (e.g., Holm) or false-discovery rate.

Response: We thank the reviewer for this comment. In the protocol we note that "No formal multiple comparisons will be employed for safety endpoints or secondary endpoints." Typically, small Phase I studies do not apply formal adjustments for multiplicity, because these studies are designed primarily to assess safety, tolerability, and immunogenicity in a descriptive and exploratory

manner, rather than to formally test hypotheses with confirmatory statistical inference. With the small sample size (10 per group), early-phase objectives, and multiple exploratory endpoints, our intent was to focus on identifying potential safety signals and the ability of this vaccine to elicit an immune response that would underlie advancing this vaccine candidate into larger Phase 1 and/or Phase II/III studies. Applying formal multiplicity corrections (e.g., controlling the family-wise error rate or false discovery rate as requested by the reviewer) would substantially reduce statistical power in this context, increasing the likelihood of missing important preliminary findings. For this reason, and as per the protocol, we did not correct for multiplicity and interpret p-values descriptively and in the context of the study's exploratory objectives. We added a statement in the Discussion section to make it clear to the reader how to view the results of this trial (line 474-476).

2.3 Complete-case analyses assume data are missing completely at random—unlikely over 56 weeks.

Response: The majority of study participants completed all study visits within the protocol-defined windows with missingness unrelated to the length of the study. Three participants had missed visits:

- One participant in 10 mcg group discontinued study product administration after the first dose based on IND sponsor/PI decision but completed follow-up visits and was included in the safety evaluation (these details are included in the Figure 1 CONSORT; line 159-160)
- One participant in 25 mcg group missed a single visit (Week 6 visit, two weeks after the second vaccination). We added these details in the Results section (lines 157-158)
- One participant in 50 mcg group moved from the area and missed Weeks 44 and 56 visits (these details are included in the Figure 1 CONSORT; lines 159).

Given this minimal and sporadic missingness which was due to personal reasons (2 out of 3 participants), rather than to study-related safety outcomes, the assumption underlying complete-case analysis is unlikely to be meaningfully violated. Furthermore, the small sample size and exploratory nature of this Phase I trial limit the feasibility and interpretability of more complex imputation methods. Therefore, complete case analysis was considered appropriate for this study.

2.4 Both intent-to-treat (ITT) and per-protocol sets are defined but only ITT results are shown. Present both or justify their merger, noting at least one participant missed dose 2.

Response: We describe ITT and per-protocol sets in the protocol; however, in the immunogenicity analysis section we state that the per-protocol analysis can be performed “if needed...” We did not perform a per protocol, analysis because this would require excluding 3 participants from our analyses: one participant who received only the first dose of vaccine; one participant who missed the Week 6 visit blood draw; and one participant who missed the Week 44 and Week 56 blood draw visits. Excluding these 3 participants would compromise the safety and immunogenicity analyses of this small study. Per protocol analyses are essential for advanced phase clinical studies in which vaccine efficacy is a main endpoint; however, in this first-in-human study our intent is to describe reactogenicity to mRNA-1215, to identify potential severe adverse events (with a group of 10, the study had over a 90% chance to observe at least 1 serious adverse events if the true event rate was at least 21%), and to describe immune responses to mRNA-1215.

We are presenting a modified intent-to-treat analysis whereby we include the data of the single 10 mcg dose participant through Week 4 given that all 40 participants had data from all timepoints up to Week 4. We believe the immune response after just one dose is an important analysis, because this provides data on how quickly someone could generate an immune response in an outbreak setting. We did not include the single dose participant in our analysis after the second dose administration at Week 4, because we wanted to analyze immune response data only from participants who had received both the first and second doses of vaccine. Our primary immunogenicity goal was to describe the magnitude and durability of the immune responses to vaccine antigens after the first and second doses of vaccine.

2.5 Time-course GMTs are plotted but not modelled. A mixed-effects model (log-titre ~ dose × time, random intercepts) should be considered to exploit within-subject correlation.

Response: We thank the reviewer for this suggestion. We have performed this analysis for the primary immunogenicity endpoint employing mixed models for repeated measures as an additional analysis for the NiV ELISA immunogenicity analysis and added the findings in the Results section (lines 271-274).

3. Results presentation

3.1 Add 95 % CIs to every figure/table (some Extended Data panels show raw counts only).

Response: Per reviewer’s suggestion, we added GMTs and 95% CIs details in the Extended Figure 4 legend and enhanced the visibility of GMTs and 95% CIs in the figure. Extended Data Tables S3, S5, S5, and S7 have 95% CIs, marked “NA” (not applicable), for the Week 0 and Week 1 timepoints (Table S3 only) because negative, baseline neutralization titers were imputed to be equal to one-half of the

lowest dilution tested which was the limit of detection of the assay and these values were the same for all participants. Additional details about GMTs and 95% CIs were included in the Statistical analysis section lines 1110-1111.

3.2 Improve figure accessibility—use distinct shapes or patterns, not just colours, to accommodate colour-blind readers.

Response: We thank the reviewer for this helpful suggestion. We have revised Figures 3, 4, 5, and Extended Data Figure 4 by not only using different colors to distinguish each dose group but also by adding distinct shapes to accommodate color-blind readers (gray circles, red squares, blue triangles, and green diamonds that represent 10, 25, 50, and 100 mcg dose groups, respectively).

4. Interpretation and generalizability

4.1 Immunogenicity is compared with convalescent sera collected > 20 years post-infection; differences in age, waning immunity and assay technology weaken that benchmark.

Response: We appreciate the reviewer raising this important point and acknowledge that, in this instance, comparison to titers of these convalescent sera is not an appropriate benchmark. High fatality rates and the sporadic nature of Nipah virus outbreaks result in very limited convalescent serum availability. For the convalescent sera used in this trial, we do not have any demographic information or details of the history of infection, including whether anyone living in an endemic area possibly had recurrent infection, to provide additional relevant context for these sera. To our knowledge, there were no attempts to measure these responses closer to the time of infection or anytime during the subsequent 20 years in these recovered patients to indicate the time course and durability of the immune responses. Therefore, although convalescent sera can serve as biologically relevant comparators for evaluating vaccine immunogenicity, we recognize that due to the very remote history of infection and lack of other information about these sera, the utility of these comparisons is limited, and hence, not scientifically informative. In consideration of the reviewers' comments, we have elected to remove vaccine-induced binding and neutralizing antibody titer comparisons to the convalescent sera titers from the manuscript.

4.2 Statements of “dose independence” should be tempered; with ten participants per group, only extreme differences are detectable.

Response: Per reviewer's suggestion, we modified the language to temper the statements regarding the ‘dose independence’ (lines 339, 374, 432) and changed it to “comparable between dose groups”.

4.3 Outline how these phase-1 data map to surrogate endpoints acceptable under accelerated or Animal-Rule regulatory pathways.

Response: For the reviewer's reference, this important topic has been addressed in a separate paragraph in the Discussion section of the manuscript, please see lines 460-471. We note that a surrogate endpoint could be established based on identifying a correlate of protection, such as a binding or neutralizing antibody response, in a non-human primate challenge model. Phase 1/ 2 studies could be designed to include such an endpoint as a surrogate one for efficacy.

5. Minor clarifications

5.1 Specify whether GMT CIs are exact or asymptotic (back-transformed log-means?).

Response: Reported GMTs and CIs are asymptotic. We added this detail in the Statistical analysis section of the Methods (lines 1110-1111).

5.2 Describe how values below the lower limit of quantification were handled.

Response: Lines 1121-1124: "For ELISA binding assays, all titer values, including those below the lower limit of quantification (LLOQ), were detected and used in the analyses. For neutralization assays, titer values that were negative were imputed as the reciprocal of half of the lowest dilution tested, which was 1:50, the limit of detection (LOD)".

Referee #2: not available

Reviewer #3 (Remarks to the Author):

Brief summary and key results:

This manuscript discusses an open-label, phase 1, first-in-human, dose-escalation study designed to evaluate the safety, tolerability, and immunogenicity of a two-dose vaccination regimen of the chimeric Pre-F/G mRNA vaccine, mRNA-1215, in healthy adult volunteers. The vaccine is notable for incorporating the stabilized pre-F antigen, representing a first-in-class approach for clinical testing.

Overall, the study showed that the vaccine was generally safe, with only mild to moderate side effects consistent with prior experience from mRNA vaccine trials.

Though, of note, a single case of prolonged urticaria occurred in the high-dose group, a side effect also observed in some recipients of COVID-19 mRNA vaccines. This reaction required antihistamine treatment for approximately five months before full resolution.

Administration of two doses, four weeks apart, induced strong antigen-specific antibody titers with demonstrated neutralizing activity. The study commendably followed titers for over a year, providing evidence for the durability of the immune response. Investigators also assessed the breadth of protection by comparing neutralization against both NiV(BM) and NiV(B) strains, as well as cross-reactive responses to HeV, evaluating both binding and neutralizing antibody titers.

General assessment:

The trial was conducted according to protocol, with only one deviation in which

one participant received a single dose, as appropriately noted in the report. The data are presented clearly, and the results are thoroughly discussed. Appropriate standards were used as a comparison when possible. Overall, I agree with the conclusion that the findings are favorable to continue the clinical development of mRNA-1215 against Nipah.

Comments:

1. Clarification on B-cell response data. I found the data somewhat confusing. In the panels Figure 4 panels a and b, the frequencies of antigen-specific memory B-cell responses were assessed at various study time points using fluorescently labeled Pre-F and G trimer probes, either matched for both NiV(M) and NiV(B) (top) or specific to NiV(M) antigens (bottom). The top panel shows a clear response for both Pre-F and G, whereas the bottom panel demonstrates no detectable response for Pre-F and only a weak signal for G. Does that mean that the NiV(B) probes account for the apparent difference? This result seems unexpected, given that the vaccine was designed based on NiV(M). Some additional clarification or interpretation from the investigators would be helpful.

Response: To clarify, the data in Figure 4 a and b show the frequencies of memory B cells that bind both Malaysia and Bangladesh probes [NiV(M)⁺ NiV(B)⁺] or only Malaysia probes [NiV(M)⁺ NiV(B)⁻]. While study participants were vaccinated with PreF/G of a NiV Malaysian strain, all the F-specific B cells can bind both the Malaysia and Bangladesh probes indicating 100% cross-reactivity (Figure 4 panel a, top and bottom). However, there is a very small portion of the G-specific B cells that only bind the Malaysia probes and not the Bangladesh probes [NiV(M)⁺ NiV(B)⁻] (Figure 4, panel b, top and bottom; see also Extended Data Fig. 1). This means that the vast majority of G-specific B cells are cross-reactive, but a small fraction can only recognize Malaysian G. Not surprisingly, there were no B cells that only bound the Bangladesh probes. As explained in the discussion (line 443-447), this is consistent with the frequency of amino acid identity between NiV(M) and NiV(B) glycoproteins (identity percentage of 98.7% for F and 95.5% for G). With such a high similarity, one would expect most B cells to be cross-reactive between the two strains. It is interesting that the ~3% difference in amino acid identity between F and G is enough to generate some G-specific B cells that can't bind the Bangladesh G protein.

Per reviewer's suggestion, we added a clarification in the results section (line 307, 321-323) and moved the following statement from the Discussion section into the Results: "identity percentage of 98.7% for F and 95.5% for G" (line 307).

2. Lack of details on vaccine formulation. The manuscript provides little information on the vaccine formulation, particularly regarding the lipid composition. I assume that the ratio of mRNA to lipid remains constant across the dose-escalation study, but does this also imply that the lipid dose increases proportionally? Since lipid nanoparticles can influence both immunogenicity and reactogenicity, I would prefer to see the authors add

more detail on the formulation as well as a brief discussion on the potential toxicity of the lipid components.

Response:

We added additional details about the lipid composition and mRNA-1215 vaccine formulation (lines 1037-1042). Since the investigational product that was evaluated in this trial consists of mRNA encapsulated in a lipid nanoparticle, and because we did not have a cohort that received an empty nanoparticle or its components, we do not think that a discussion of the potential toxicity of the lipid components is warranted in this report of our trial results.

Reviewer #4 (Remarks to the Author):

In this manuscript authors analyzed vaccine safety, tolerability and immunogenicity of the lipid nanoparticle mRNA vaccine against Nipah virus (NiV) in the phase 1, first-in-human, open-label, dose-escalation trial. They immunized intramuscularly 40 volunteers, with mRNA-1215, using 2 doses at a 4-week interval. They observed in immunized individuals reactogenicity symptoms including mild pain and tenderness at the injection site (n=33) and mild malaise (n=16). mRNA-1215 elicited robust NiV F and G binding antibody titers and neutralizing titers by 2 weeks after the prime, remaining detectable elevated for at least one year after vaccination. They suggest mRNA-1215 as a promising vaccine candidate to advance for pandemic preparedness and for those at risk from regional outbreaks. Nipah virus is highly pathogenic virus for which neither licensed vaccine nor therapeutics are currently available, underlying the importance of the development of novel prophylactic approaches, like the one presented in this manuscript.

1. The study of vaccine safety and tolerability was well performed and results presented correctly. However, all studies of vaccine immunogenicity were performed without any test implicating the live virus. Although the utilization of pseudovirus seroneutralisation assay most often correlates with the neutralization of the live virus, it cannot completely replace it in the clinical trial of the new Nipah vaccine and is indispensable to claim the new vaccine candidate for Nipah. This is even more important as the proof-of-concept study of the mRNA-1215 vaccine in nonhuman primates is still not published, making it difficult to verify results presented only in the meeting abstract (ref. 35 of the manuscript).

Response: Pseudovirus neutralization assays are widely used in early-stage vaccine studies to assess immune responses and are especially useful when dealing with a highly lethal pathogen (Nipah virus is a biosafety level 4 pathogen). For initial licensure or emergency use authorization of a vaccine, however, live virus neutralization assays are required by regulatory agencies, such as the FDA and EMA. Therefore, as this vaccine advances further into clinical development, live virus neutralization assays may be employed and compared to pseudovirus-based neutralization assays to determine if they correlate. This question will also be addressed in pre-clinical Nipah challenge studies, which will soon be

submitted. In any case, we added this as one of the study limitations (lines 481-486).

2. Authors claim that “starting 2 weeks after vaccination, the neutralizing antibody titers induced by mRNA-1215 were similar to titers of NiV convalescent patients“(line 218, Fig. 3f). As the sera from convalescent patients were obtained more than 20 years after NiV infection and the antibody titer wanes with the time, this suggest that mRNA-1215-induced neutralizing titer is not very high, which does not seem to be particularly rewarding as a message of the paper?

Response: Response: We appreciate the reviewer raising this important point and acknowledge that, in this instance, comparison to titers of these convalescent sera is not an appropriate benchmark. High fatality rates and sporadic nature of Nipah virus outbreaks result in very limited convalescent serum availability. For the convalescent sera used in this trial, we do not have any demographic information or details of the history of infection, including whether anyone living in an endemic area possibly had recurrent infection, to provide additional relevant context for these sera. To our knowledge, there were no attempts to measure these responses closer to the time of infection or anytime during the subsequent 20 years in these recovered patients to indicate the time course and durability of the immune responses. Therefore, although convalescent sera can serve as biologically relevant comparators for evaluating vaccine immunogenicity, we recognize that due to the very remote history of infection and lack of other information about these sera, the utility of these comparisons is limited, and hence, not scientifically informative. In consideration of the reviewers’ comments, we have elected to remove vaccine induced binding and neutralizing antibody titer comparisons to the convalescent sera titers from the manuscript.

3. In the discussion (line 302), authors claim that they are no other published vaccines which target both NiV F and G. The statement should be corrected as there are others published work presenting this type of vaccines (for ex. Pastor Y. et al, 2024).

Response: We thank the reviewer for bringing the study by Pastor *et. al* titled “A vaccine targeting antigen-presenting cells through CD40 induces protective immunity against Nipah disease” to our attention. We have now included a reference to this work in the discussion section alongside other Nipah virus vaccines that target both NiV F and G antigens that were evaluated in preclinical studies (line 400). As the suggested study was conducted in African green monkeys (AGMs), we maintain that to our knowledge Nipah virus vaccine, mRNA-1215, is the only vaccine that includes both F and G antigens and that has been evaluated in a human clinical trial. A qualification was added to the text noting this fact (line 400-401).

4. How do authors explain that CD8 T cell responses were generated following vaccination only against NiV F and not NiV G?

Response: The CD8 T cell responses to both NiV F and G were very low as previously seen following mRNA vaccination against SARS-CoV-2 (Anderson. E.J. *et al* NEJM, 2020;doi: 10.1056/NEJMoa2028436). While nearly undetectable, there is a trace of a CD8 T cell response to NiV G. The lower CD8 T cell response to G is consistent with the overall lower CD4 Th1 response to G as shown in Extended data figure 4, panel a. Further, F protein is larger than G and therefore has a larger number of peptides in its peptide pool (123 versus 108 individual peptides, respectively), thus allowing for the presentation of more epitopes. Finally, it's possible that F is intrinsically more immunogenic than G. A minor edit was added to the discussion (line 378).

Editorial Requests:

In terms of formatting, please include a summary safety data (like Table S2) and the baseline characteristics (Table S1) as main text figures/tables.

Response: We have included “Demographic Characteristics of Study Participants” as Table 1 and “Unsolicited Adverse Events Related to mRNA-1215 Vaccination” as Table 2 in the main text figures/tables.

Please provide exact dates (M,D,Y) for first and last patient enrollment in the main text (right now it is noted in a figure legend).

Response: These details are included in the Results: Trials participants section line 148.

Please also update your data availability statement. It must indicate any restrictions on data/code availability.

Response: This statement has been updated lines: 1274-1289.

Please indicate whether individual-level data will be made available, when, and for how long.

Response: We added the details are requested lines 1279.

Please provide a transparent path detailing how external requests will be evaluated, including contact information and an estimated timeframe for response when applicable.

Response: We added these details as requested line 1278-1286.

Finally, please also review our SAGER guidance, and discuss whether and sex-based analyses were conducted here (it is not required to do this post-hoc, we would just like transparent reporting about whether this was pre-planned).

Response: We indicated in the Methods section that no sex-based analysis has been performed in this trial line 1117-1119.

Please also provide more information about sex/gender and race/ethnicity reporting in the methods (how participants self-reported sex race and ethnicity, what options were given, and please also clarify if gender information was collected).

Response: We included these details in the Study Procedures section lines 1058-1062. Participants were provided a demographic form to fill out after enrollment and indicate their sex, race, and ethnicity. The original form asked about gender identify. We have not provided this information or data in the manuscript because of President Trump’s Executive Order that was issued on the day of his inauguration, January 20, 2025. <https://www.whitehouse.gov/presidential-actions/2025/01/defending-women-from-gender-ideology-extremism-and-restoring-biological-truth-to-the-federal-government/>. Given that this study was conducted with U.S federal dollars at the U.S. National Institutes of Health, we cannot use gender information or data in the manuscript, because it is prohibited

by this Executive Order.

Please edit your manuscript to comply with our formatting guidelines for Articles, which are:

Abstract: 200-300 (maximum) words, the abstract must indicate the phase of the trial: signpost the primary and secondary/exploratory endpoint data summarized in this section, including safety.

Response: We added these details as requested: line 54, lines 57-58.

You must state if the primary outcome was or was not met (and use the terminology "met pre-specified endpoints") and provide the effect size and relevant uncertainty estimate.

Response: We clarified that “the primary outcome of vaccine safety and tolerability met pre-specified endpoints (line 54).” We did not provide the effect size and relative uncertainty estimate because this smaller vaccine trial is first-in-human, descriptive, and hypothesis generating study of a unique, structure-based vaccine. The analysis for the primary safety outcome was pre-determined to be descriptive, summarizing the number and percentage of participants who experienced reactogenicity and adverse events after vaccination as noted in Section 6, Statistical Considerations, of the protocol; it was not powered to detect a ‘treatment effect’ between the groups, as is done for larger , advanced stage clinical trials that also include a control group.

The paragraph must conclude with the trial identifier in the following format: ClinicalTrials.gov identifier: NCTXXX [hyperlinked]

Response: We added this information to the abstract: line 62.

Please also include the number of participants in each arm of the trial

Responses: We added this in line 52.

* Main text: 4000-5000 (maximum) words with subheadings for the Introduction, Results and Discussion. The Results should start with a trial design subsection that describes key eligibility criteria and lists all endpoints in line with what was prespecified in the protocol flagging any not reported in this manuscript. We prefer the results section to be:

- Patient disposition
- Primary outcome(s)

- Secondary outcomes
- Safety
- Exploratory outcomes
- Sensitivity analyses
- Post-hoc analyses

Response: We added the key eligibility criteria to the beginning of the results section (lines 149-155) and listed all the pre-specified safety endpoints in the vaccine safety paragraph (line 165-173). We did not rename the results subsection headings, as noted above, but our result section headings do present the content as outlined above: Trial Participants (includes Patient Disposition); Primary outcome (Vaccine Safety); Secondary Outcome (Vaccine-induced binding antibody responses to NiV(M) Pre-F and G; Exploratory Outcomes [Neutralizing antibody responses to NiV(M), B cell responses to NiV..., T cell responses against NiV (M), Cross-reactive binding and neutralizing antibody responses to HeV]. In the beginning of each section, we indicate which type of data are presented (line 255-256, 292-293, 305-306, 325-327, 334-335, 348-350). In addition, we included a post-hoc model sensitivity analysis for the secondary endpoint (271-274).

The Methods should include a full description of the inclusion and exclusion criteria, as well as study procedures and statistical analyses (including a power calculation). Please note that our methods section does not have a word limit.

Response: We added a full description of the inclusion and exclusion criteria lines: 958-1010; study procedures are detailed lines: 1045-1087; and addition details and clarifications to the statistical analysis section based on Reviewer #1's overall comments: lines: 1099-1146.

References: approximately 60 in the main text + 20 methods-only references

Response: We have provided references separately for the Main text and Methods.

Display items: up to 6 main display items (inclusive of figures and tables) and up to 10 Extended Data display items (inclusive of figures and tables). Extended Data are an integral part of the paper and only data that directly contribute to the main message should be presented. Note that patient baseline characteristics, primary endpoint data and the CONSORT-style trial flow diagram must all be main display items.

Response: We moved baseline characteristics table into the main text. CONSORT diagram and primary endpoint data are displayed in the main text.

The Introduction should be written for a broad, non-specialist medical reader and provide sufficient context for the work. Please ensure all results are presented in the Results section, no new data should be introduced in the Discussion.

Response: We complied with this requirement.

You must include explicit paragraphs of study limitations in the Discussion.

Response: Study limitations are addressed in the Discussion section lines: 474-486.

The overarching conclusion of the study must be based only on the primary outcome and safety data.

Response: The overarching conclusion of the trial related to the primary safety outcome is detailed in the first and second paragraphs of the Discussion section lines: 362-396.

Please upload the protocol and SAP with the revision materials, so that reviewers and editors have access to them.

Response: We submitted the study protocol along with the manuscript files. Since this was a small, first-in-human Phase 1 trial, a separate SAP was not developed or required by the ethics committee or by the U.S. FDA. This trial was reviewed and approved by FDA and NIAID IRB without requiring a SAP. The analyses performed in this trial are outlined in Section 6 of the protocol and were aligned with protocol objectives of safety, tolerability, and immunogenicity and were primarily descriptive and hypothesis generating.

You must ensure that contributions from all individuals in the author list are available in the Author Contributions statement.

Response: Author contributions included in the manuscript lines 540-548.

Please move all funding sources to the Acknowledgements, including a statement on the role of the funder. Please acknowledge who funded the study in the first sentence of the acknowledgements.

Response: The Acknowledgement section has been modified as requested lines 507-538.

Please ensure that all potential competing interests are detailed for all authors. For any authors with no competing interests, this must also be stated.

Response: Competing interest can be found in the manuscript lines: 550-558.

The article file must only contain these items in this order: • Title

- Author List and affiliations
- Abstract
- Introduction
- Results (with Subheadings)
- Discussion
- Acknowledgements
- Author Contributions
- Competing Interests Statement
- References (for main text only)
- Figure legends (for main text only)
- Tables (note: tables should be pasted into Word files as editable tables, not as images)
- Methods
- Data Availability Statement
- Code Availability Statement
- Methods-only References

Response: We organized the sections of the manuscript according to the guidelines provided.

We therefore invite you to revise your manuscript taking into account all reviewer and editor comments. Please highlight all changes in the manuscript text file.

Response: We provided a manuscript with all changes tracked.

Include a point-by-point response to all editorial queries below. For any points that do not apply to your revision, please state “not applicable.”

Response: We provided point-by-point response to all editorial queries.

Include a “Response to referees” document detailing, point-by-point, how you addressed each referee comment. If no action was taken to address a point, you must provide a compelling argument. This response will be sent back to the referees along with the revised manuscript.

Response: We have included point-by-point response to all reviewers’ comments.

If you have not already done so, please begin to revise your manuscript so that it conforms to our policy and format instructions here: <https://www.nature.com/nm/for-authors/preparing-your-submission#formatting> Refer also to any guidelines provided in this letter.

Submit both clean and tracked changes versions of the revision as editable Word documents.

Response: We submitted both clean and tracked versions of the manuscript as editable Word documents.

Include a revised version of any required reporting checklist. The reporting summary can be found at <https://www.nature.com/documents/nr-reporting-summary.pdf>. Please note that this form is a dynamic ‘smart pdf’ that must be downloaded and completed in Adobe Reader (i.e. cannot be opened in a web browser).

Response: We have updated the Reporting Summary checklist to include mmmr package in R which was used to perform sensitivity analysis.

Nature journals have recently announced an update to our guidance on reporting on sex and gender in research studies (see here). We strongly encourage researchers to follow the ‘Sex and Gender Equity in Research – SAGER – guidelines’ and to include sex and gender considerations for studies involving humans, vertebrate animals and cell lines where relevant to the topic of study (an overview can be found here). Authors should use the terms sex (biological attribute) and gender (shaped by social and cultural circumstances) carefully in order to avoid confusing both terms.

Response: In this trial, study participants only self-reported sex at birth (line 1058; within the footnotes of the Table 1).

When preparing your revised manuscript, please be aware of our guidance on Sex and Gender reporting). Please note that:

1. If the research findings apply to only one sex or gender, that must be indicated in the title and/or abstract.

- 2a. For studies involving vertebrates animal and cell lines- The Reporting Summary should include whether sex was considered in the study design.
 - 2b. For studies involving human research participants- The Reporting Summary should include whether sex and/or gender was considered in the study design and whether sex and/or gender of participants was determined based on self-report or assigned (and methodology used).
 3. Data should be reported disaggregated for sex and gender where this information has been collected and consent has been obtained for reporting and sharing individual-level data; disaggregated numbers for individual experiments must be provided in the source data as appropriate whereas overall numbers may be provided in the Nature Portfolio Reporting Summary.
- Information on the 3 points above should be included in the revised manuscript and detailed in the cover letter.

Response: No sex-based analyses were performed as this was not an objective of the trial. Such investigation is beyond the scope of this small phase 1 clinical trial and should be further investigated in the subsequent clinical investigation phases. We collected gender data but are not able to include these data due to President Trump's Executive Order <https://www.whitehouse.gov/presidential-actions/2025/01/defending-women-from-gender-ideology-extremism-and-restoring-biological-truth-to-the-federal-government/> Given that this study was conducted with U.S federal dollars at the U.S. National Institutes of Health, we cannot use gender information or data in the manuscript, because it is prohibited by this Executive Order.

In addition, please note that if sex- and gender-based analyses have been performed a priori, results should be reported regardless of positive or negative outcome. We discourage conducting post hoc sex- and gender-based analysis if the study design is insufficient (for example, low sample size) to enable meaningful conclusions. If no sex- and gender-based analyses have been performed, please indicate the reasons for the lack of these analyses in the Reporting Summary.

Response: No sex-based analyses were performed as this was not an objective of the trial (line 1117-1119). Such investigation is beyond the scope of this small phase 1 clinical trial and should be further investigated in the subsequent clinical investigation phases. We have indicated this in the Reporting Summary.

When submitting the revised version of your manuscript, please also pay close attention to our href="<https://www.nature.com/nature-research/editorial-policies/image-integrity>">Digital Image Integrity Guidelines. and to the following points below:
-- that unprocessed scans are clearly labelled and match the gels and western blots

presented in figures.

-- that control panels for gels and western blots are appropriately described as loading on sampleprocessing controls

-- all images in the paper are checked for duplication of panels and for splicing of gel lanes.

Reviewer #1 (Remarks to the Author):

Thank you for revision. I have no further comment.

Response: We thank the reviewer for the positive feedback and for the constructive comments that helped improve our manuscript.

Reviewer #3 (Remarks to the Author):

I am satisfied with the changes made by the authors in response to my comments. In addition, I found the additional changes made based on the suggestions of the other reviewers to further clarify the data of the clinical study.

Response: We are grateful to the reviewer and appreciate learning that the revisions were helpful and improved the clarity of the manuscript.

Reviewer #4 (Remarks to the Author):

Authors have successfully answered to this reviewer's comments (although the lines marked to contain the modifications to responses to questions 1 and 4 were different in the discussion of the modified manuscript).

Response: We thank the reviewer for the comments and suggestions. We apologize for the incorrect line references, which may have resulted from manuscript reformatting.